# Structure of the N-RNA/P interface indicates mode of L/P recruitment to the nucleocapsid of human metapneumovirus

Jack D. Whitehead [1,2], Hortense Decool [3], Cédric Leyrat [4], Loic Carrique [1], Jenna Fix[3], Jean-François Eléouët[3], Marie Galloux[3] ✉ & Max Renner [5,6] ✉

Human metapneumovirus (HMPV) is a major cause of respiratory illness in young children. The HMPV polymerase (L) binds an obligate cofactor, the phosphoprotein (P). During replication and transcription, the L/P complex traverses the viral RNA genome, which is encapsidated within nucleoproteins (N). An essential interaction between N and a C-terminal region of P tethers the L/P polymerase to the template. This N-P interaction is also involved in the formation of cytoplasmic viral factories in infected cells, called inclusion bodies. To define how the polymerase component P recognizes N-encapsidated RNA (N-RNA) we employed cryogenic electron microscopy (cryo-EM) and molecular dynamics simulations, coupled to activity assays and imaging of inclusion bodies in cells. We report a 2.9 Å resolution structure of a triple-complex between multimeric N, bound to both RNA and the C-terminal region of P. Furthermore, we also present cryo-EM structures of assembled N in different oligomeric states, highlighting the plasticity of N. Combined with our functional assays, these structural data delineate in molecular detail how P attaches to N-RNA whilst retaining substantial conformational dynamics. Moreover, the N-RNA-P triple complex structure provides a molecular blueprint for the design of therapeutics to potentially disrupt the attachment of L/P to its template.

Human Metapneumovirus (HMPV) was first reported in 2001 when it was isolated from children in the Netherlands with symptoms ranging from mild respiratory disease to severe pneumonia[1]. HMPV is thought to have a zoonotic origin following a spillover event from an avian reservoir host species[2]. The pathogen is now known to cause both upper and lower respiratory tract infections, including both bronchiolitis and pneumonia, with children being the most affected demographic[3–6]. It is estimated that 5% to 12% of hospitalizations due to viral respiratory infections in children are caused by HMPV[3,7]. Furthermore, infections with HMPV can also be severe in the elderly and in

patients with comorbidities[8]. Serology studies carried out in 2001 found that by the age of 5, virtually all children had been exposed to the virus[1]. While vaccines against the closely-related respiratory syncytial virus (RSV) have been recently approved for use in pregnant individuals and the elderly[9–12], there are currently no licensed vaccines or specific therapeutics for the treatment of HMPV infections. A better understanding of the molecular mechanisms of the HMPV life cycle are therefore needed.

HMPV shares many features with RSV, both being non-segmented negative-sense RNA viruses (nsNSVs) belonging to the *Pneumoviridae*

[1]Division of Structural Biology, The Wellcome Centre for Human Genetics, University of Oxford, Oxford, UK. [2]Sir William Dunn School of Pathology, University of Oxford, Oxford, UK. [3]Université Paris-Saclay, INRAE, UVSQ, VIM, 78350 Jouy-en-Josas, France. [4]Institut de Génomique Fonctionnelle, Université de Montpellier, CNRS, INSERM, Montpellier, France. [5]Department of Chemistry, Umeå University, Umeå, Sweden. [6]Umeå Centre for Microbial Research, Umeå University, Umeå, Sweden. ✉e-mail: marie.galloux@inrae.fr; max.renner@umu.se

family[13]. HMPV possesses a 13.3 kb negative-sense RNA genome containing eight genes: 3′-N-P-M-F-M2-SH-G-L-5′[14]. The F, SH, and G proteins constitute surface glycoproteins[15–17]. The M and M2 genes encode the matrix protein M, and the M2-1 and M2-2 proteins via overlapping ORFs[18–20]. The L protein harbors the RNA-dependent RNA polymerase activity (RdRP) and is responsible for the transcription of capped and polyadenylated viral mRNAs as well as replication of the genome[21]. The viral genome is encapsidated in a sheath of oligomerized copies of the nucleoprotein N, forming a ribonucleoprotein complex termed nucleocapsid[22,23]. The nucleocapsid is the template for transcription and replication by L, which is also dependent on the obligate polymerase cofactor, the phosphoprotein P, together forming the active L/P holoenzyme[24].

A hallmark of nsNSVs is the formation of specialized viro-induced cytoplasmic inclusions which concentrate viral RNA and proteins[25]. During RSV and HMPV infections, these inclusion bodies (IBs) have been shown to harbor L/P and N and to be active sites of viral transcription and replication[26,27]. *Pneumoviridae* IBs are spherical membrane-less organelles. Recent studies provide evidence that IBs are biomolecular condensates that form through liquid-liquid phase separation (LLPS)[28–30]. HMPV IBs mature in the course of infection and grow via actin-dependent coalescence and fusion of replicative sites[27]. In a cellular context, the minimal elements for the morphogenesis of pneumoviral condensates are the P and N proteins, which together can form pseudo-IBs, even in the absence of any other viral proteins[29]. The interaction between N and P has been previously demonstrated to be crucial for the formation of cellular IBs[31,32]. However, the HMPV P protein is also able to form liquid condensates through LLPS in the absence of N, at least in vitro[32].

The nucleoprotein N possesses globular N-terminal and C-terminal domains (NTD and CTD, respectively)[22,33]. In oligomerized and RNA-bound N (N-RNA) the nucleic acid is threaded along an extended groove in between the NTDs and CTDs of laterally connected N protomers. Furthermore, N possesses short N-terminal and C-terminal extensions (termed arms) which contact neighboring N protomers and facilitate oligomerization. In the context of infected cells, nucleocapsids are co-transcriptionally assembled from $N_0$-P – RNA-free N protomers which are kept in a monomeric state by binding to the N-terminal region of P. A crystal structure of the HMPV $N_0$-P complex has been previously reported[33]. The HMPV P protein is modularly disordered and forms a tetramer via a central coiled-coil domain[34,35]. Upstream and downstream of the coiled coil, P possesses extended regions which are mainly intrinsically disordered or present transient secondary structures in the absence of binding partners, but can conditionally fold upon interaction with other proteins. For instance, a recent cryo-EM reconstruction of the HMPV L/P complex showed contextual folding of regions of P lying downstream of the coiled coil through interaction with the polymerase[24]. However, in the same structure, only 22% of the total residues of tetrameric P present in the construct were visible in the cryo-EM density, with the remainder being disordered. These substantial conformational dynamics make structural elucidation projects involving P challenging.

An important function of the C-terminal domain of P in nsNSVs is to bind to assembled N-RNA, forming an N-RNA/P complex. However, the nature of the N-RNA/P interaction differs within nsNSV families. In the family *Paramyxoviridae* (e.g. measles virus), N proteins possess a long (up to ~170 residues) unstructured tail termed $N_{tail}$, which extends far from the nucleocapsid[36–38]. A region of the paramyxoviral $N_{tail}$ interacts with the so-called $P_{XD}$ domain of the phosphoprotein, forming a $P_{XD}$-$N_{MoRE}$ complex[39–42]. Recently, analysis of compensatory mutations after N truncation has indicated that the $P_{XD}$-$N_{MoRE}$ complex can also attach to the core of N, however high-resolution structural data of this interaction is not yet available[43]. In the family *Rhabdoviridae*, which includes rabies and vesicular stomatitis viruses, the C-terminal domain of the P protein is less disordered than in other

nsNSVs and attaches to two C-terminal domains of neighboring N-RNA protomers[44,45]. Finally, in the *Pneumoviridae* HMPV and RSV the N-RNA/P attachment occurs via an interaction of the disordered C-terminal region of P ($P_{CT}$) with N[31,32,46]. The N-RNA/P interaction is thought to be important in multiple steps of the viral replicative cycle: 1) tethering of the L/P complex to the nucleocapsid during transcription/replication, 2) formation of IBs, 3) loading of nucleocapsids into assembling progeny virions through an interaction with the viral matrix[47]. Even though these constitute crucial functions, the structural basis of this interaction in HMPV has thus far remained elusive.

In this study we report cryo-EM structures of recombinant HMPV N-RNA bound to $P_{CT}$ (N-RNA/P), revealing the structural basis of the interaction surface. P binds into a groove at the periphery of the NTD of N, between a helix and an extended loop of a beta-hairpin of N (here denoted as $N_{βHL}$). We observed that the density of $N_{βHL}$ becomes more ordered upon P binding, indicative of a folding-upon-binding event. All-atom explicit solvent MD simulations of an N-RNA/P 5-mer indicated substantial conformational dynamics of the $P_{CT}$, in line with the notion of a transient interaction. We further investigated pseudo-IB formation via a functional minigenome assay to quantify the effects of mutations within the beta-hairpin loop. Our experiments confirmed that $N_{βHL}$ is important for the binding of P and that perturbations of this interaction disrupt polymerase activity and IB formation. Taken together our data elucidate the structural mechanism of P attachment to N-RNA which retains substantial conformational flexibility in the bound state. Our observations are in line with a dynamic N-RNA/P interaction which allows rapid binding and unbinding during viral transcription/replication, yet is still sufficiently strong to maintain association through the multivalent character of the interaction partners.

## Results

### Cryo-EM reveals plasticity of N-RNA assemblies

*Pneumoviridae* nucleocapsids are helical assemblies of N bound to the viral RNA genome. Our strategy to elucidate the interaction between assembled N-RNA with the polymerase cofactor P was to employ a tractable system of recombinant N-RNA rings, described previously[33]. Viral N-RNA rings are present in both infected cells as well as in virions, however, their function is currently unclear[48,49]. We reasoned that recombinant N-RNA rings could be leveraged as a tool to obtain high-resolution cryo-EM structures of a complex with P, circumventing confounding factors such as the inherent flexibility of helical N assemblies in *Pneumoviridae*[22]. As an initial step, we set out to determine the suitability of N-RNA rings for high-resolution single particle cryo-EM. HMPV N (strain NL/00/1) was expressed in *E. coli* and purified via size exclusion chromatography (Fig. 1A), followed by vitrification for cryo-EM. Upon recombinant expression, N is known to take up host RNA and self-assemble into oligomers[33]. Raw EM images suggest a heterogenous mixture of multiple oligomeric species. Template-free 2D-classification of particles revealed that the sample contains primarily rings with 10 protomers (Fig. 1B). However, 11-mer oligomers (black arrow, Fig. 1B), 12-mer oligomers (white arrow, Fig. 1B), and short coil-like segments, denoted as spiral particles, are also present in the preparation (red arrow, Fig. 1B). Similar spiral particles were observed previously in preparations of Nipah virus N-RNA[50]. We could not detect classes resembling the so-called clam particles described for RSV, Sendai or Newcastle viruses[23,51,52].

To determine if structures of suitable resolutions can be obtained from the heterogenous sample, we turned to single particle 3D reconstruction (Supplementary Table 1 and Supplementary Fig. 1A). We obtained a 3.1 Å resolution map of 10-mer rings of N-RNA (Fig. 1C and Supplementary Fig. 1B), a significant improvement from the previously reported crystallographic 10-mer at 4.2 Å[33]. We observed that the overall map quality was slightly better in the CTD region of N than in the NTD region, hinting at a degree of conformational heterogeneity

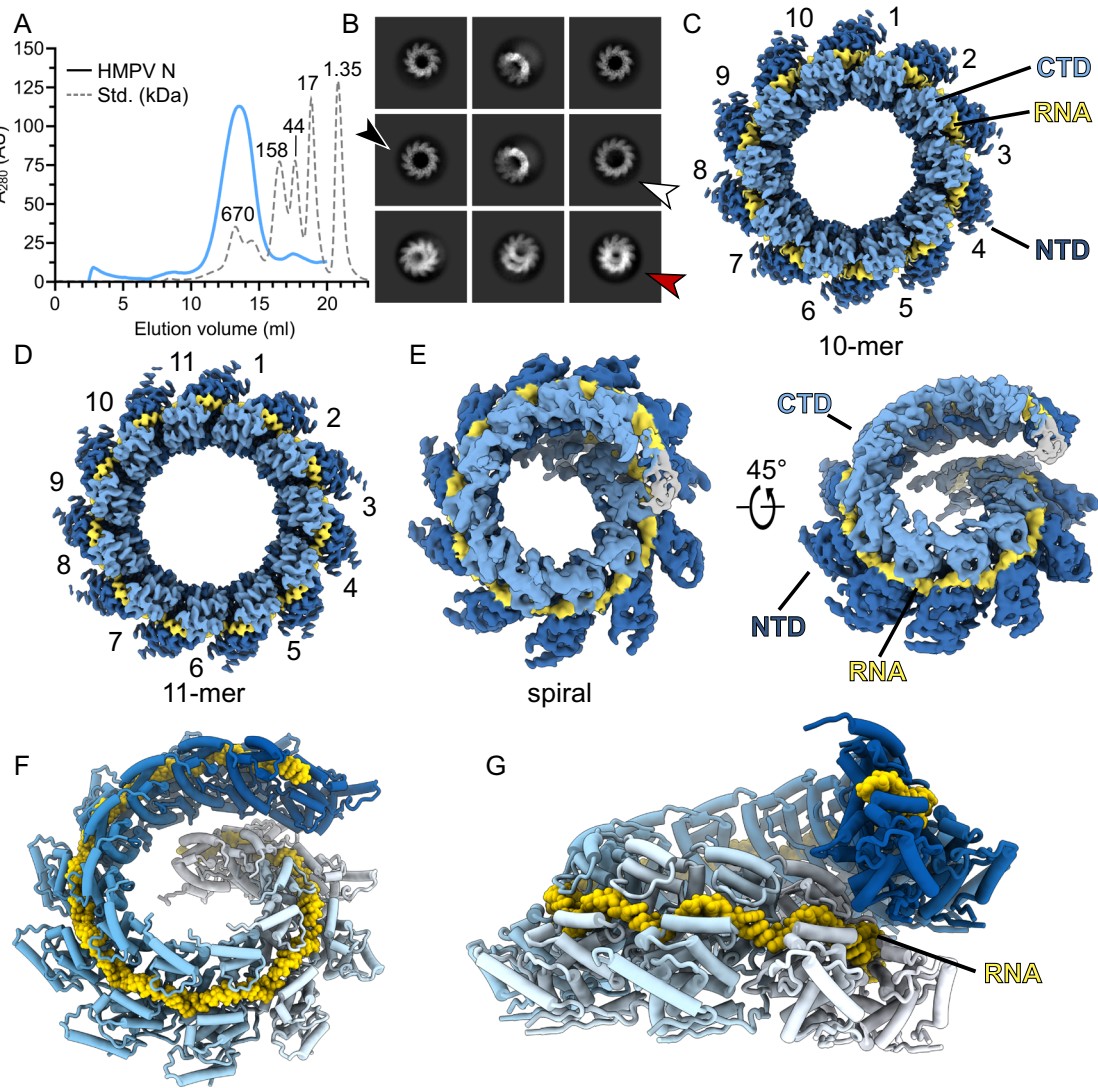

**Fig. 1 | Cryo-EM of N-RNA from HMPV. A** Size-exclusion chromatogram of recombinant HMPV N-RNA purified over a Superose 6 column. A protein standard is plotted as dotted line for comparison. **B** Representative 2D-class averages from cryo-EM. Particles were found predominantly as 10-mer oligomers. However, 11-mers (black arrow), 12-mers (white arrow), and spiral particles (red arrow) were also observed. **C**–**E** Cryo-EM maps of reconstructions of the indicated N-RNA oligomers. The bound RNA, and the C-terminal and N-terminal domains of N are labeled (CTD and NTD, respectively). The RNA is colored in yellow, the CTD in light blue, and the NTD in dark blue. **F** Tilted view and **G** side view of rigid body fitted N-RNA protomer models of the spiral assembly. N is shown in a cartoon representation, RNA is displayed as yellow spheres. The N protomers are colored sequentially from white to blue. Source data are provided as a Source Data file.

in the NTD. we also determined a map of the 11-mer ring at 3.3 Å resolution (Fig. 1D). A structural alignment of the refined N protomer models solved by cryo-EM and crystallography shows that these are highly similar (Supplementary Fig. 2A, r.m.s.d. of 0.5 Å). The 11-mer of N-RNA possesses a diameter of ~188 Å, slightly larger than the 10-mer at ~180 Å (Supplementary Fig. 2B). The ring expansion by one protomer is accompanied by a slightly increased angle between neighboring N copies in the 11-mer (Supplementary Fig. 2C): 73.6° in the 11-mer vs 72.0° in the 10-mer, in respect to the center of the rings. In the case of the 12-mer the dataset did not contain a sufficient number of side-views to obtain a reliable map. Finally, 3D-reconstruction of the spiral particles yielded a coulomb potential map at an intermediate resolution of 4.7 Å (Fig. 1E).

The map density of the spiral particle shows N protomers forming a spiral staircase and stacking up upon each other, in a mode reminiscent of a helical assembly. The map is most reliable in the central protomers and becomes increasingly disordered towards the outer protomers. As the map quality was globally too limited for atomic refinement we restricted ourselves to rigid-body placement of N-RNA

protomers. We were able to rigid body fit 12 protomers of N into the map (Fig. 1F). Thresholding of the map at a low level indicates the presence of additional N protomers, however, this density was too diffuse to fit more copies of N, presumably due to heterogeneity of the sample or conformational heterogeneity. It is notable that in this arrangement the RNA strand is obscured by N protomers belonging to the next turn of the assembly (Fig. 1G), suggesting that a conformational change is necessary for the genome to become accessible to L/P. Taking together our cryo-EM reconstructions, we observe that the lateral interaction of RNA-bound N protomers can support a range of geometries, demonstrating the plasticity of N-RNA and supporting the notion of a highly flexible HMPV nucleocapsid.

## N protomers within assemblies are laterally hooked together via a loop insertion

While the previous analysis demonstrated that N has considerable flexibility in the types of N-RNA assemblies it can form, we wanted to also explore the dynamics of N protomers within an assembly. Substantial variations in N protomer orientations in assembled N-RNA

have been observed previously in *Rhabdoviridae*[53,54]. To analyze this in *Pneumoviridae*, we turned to 3D Variability Analysis[55], a methodology that can capture a family of related structures present in the experimental data. We decided to apply this to two neighboring N protomers of a 10-mer, following 5-fold symmetry expansion and local refinement. We observed that local refinement of an N-RNA dimer yielded higher quality maps (Fig. 2A) as evidenced by the improved anisotropic displacement parameters (ADPs) of the model (Supplementary Table 1), especially in the NTD region. Variability analysis of the dimer revealed a notable motion, especially of the NTD, given by a marked transverse tilting movement of N by ~4°, with respect to the center of mass of the protomer (Supplementary Movie 1, Fig. 2B). The tilting motion occurs around a pivot point between NTD and CTD of N. As a consequence of the motion, the NTDs globally move away from each other by ~2 Å (maximum displacement up to ~5 Å at the periphery of the NTD) at the endpoint vs. the starting point of the variability component.

The intra-assembly tilting of N begs the question of which N-N interactions at the lateral flanks remain constant and support side-to-side binding throughout the motion. While the importance of the N-terminal and C-terminal arms of N has been well established in this role[33], we observed an additional mechanism of lateral N-N interaction. In all frames of the variability component, the loop ranging from residues 232-239 of the $N_i$ protomer inserts into the neighboring $N_{i-1}$ protomer (red loop in Fig. 2C). The loop of N is hooked into position from the top and bottom by residues P308 and R27 from the $N_{i-1}$ protomer, with residues S243 and E241 providing additional stabilization through H-bonding (Fig. 2D). Multiple sequence alignment (MSA) of *Pneumoviridae* N proteins reveals that the proline, arginine, and glutamate residues at these positions are conserved (black arrows in Fig. 2E). A structural comparison with the prototypical *Paramyxoviridae* member measles virus (MeV)[56,57] indicates that a similar loop is also present in other viral families (Supplementary Fig. 3). However, the involvement in N-N association may be less pronounced than in *Pneumoviridae*, based on the interacting surface areas buried by the loop (680 Å² vs only 395 Å² in HMPV and MeV, respectively)[58].

To probe the importance of these residues for the replication/transcription of HMPV we employed a minigenome assay with a luciferase-based readout of viral polymerase activity[31]. Mutagenesis of P308 to alanine resulted in a notable decrease of polymerase activity, while the R27A mutation nearly abrogated polymerase activity (Fig. 2F), suggesting that the N-N interaction via the 232-239 loop may be involved in controlling nucleocapsid assembly. Alternatively, the mutations might cause conformational changes in N that interfere with the interaction with L/P. Comparable expression levels of wild-type and mutant N proteins were observed via Western blot throughout the minigenome assays (Supplementary Fig. 4). In summary, our data point towards a critical lateral interaction between N protomers via insertion of the loop from residues 232-239, which stays consistent throughout the motion of assembled N protomers. Our data also suggest that residues P308 and especially R27 play a key role in this lateral interaction.

## The $P_{CT}$ binds to the periphery of the NTD of N

Next, we set out to resolve the interface between the C-terminal region of P ($P_{CT}$) and N-RNA. P is a flexible protein with extensive intrinsically disordered regions, and the N-RNA/P interaction has been suggested to be transient, complicating structural characterization[35,46,59]. Using the deep-learning-based Metapredict server[60], we generated disorder-score plots for both HMPV N and P. In agreement with previous work, N disorder scores are globally low, while P contains significantly disordered regions (Fig. 3A). The disorder prediction of the very C-terminal stretch of HMPV P lies in between ordered and disordered, suggestive of conditional disorder (black arrow in P disorder plot, Fig. 3A). Given the above considerations, our strategy was to shift the

equilibrium to a P-bound state of N-RNA by adding a large molar excess of a $P_{CT}$ peptide. We reasoned that the loss of contrast in the cryo-EM images stemming from unbound peptide background would not hinder us in high resolution 3D reconstruction due to the prominent and easy-to-align shape of N-RNA rings.

We incubated N-RNA preparations with a ~100-fold molar excess of synthetic peptide covering the 9 last residues of $P_{CT}$ (N-EDDIYQLIM-C). Cryo-EM analysis of the N-RNA/P complex revealed the presence of the same classes of assemblies as was the case with the apo sample. We carried out 3D reconstruction of 10-mers, 11-mers, and spiral particles of the N-RNA with $P_{CT}$ (Supplementary Table 1, Supplementary Fig. 5). In the reconstructions, additional density was visible at the periphery of the NTD of N. To facilitate easier model building and to obtain improved maps, we carried out symmetry expansion and local refinement of a dimer of N-RNA/P as before with N-RNA. The final resolution was at 2.9 Å and comparison of the N-RNA and N-RNA/P dimer maps (Fig. 3B) revealed a density of sufficient quality to allow refinement of a model of the bound $P_{CT}$ (Fig. 3C). In the case of the spiral particle, we restricted ourselves to rigid-body placement of N-RNA/P protomers due to mediocre map quality, as before with N-RNA.

The locally refined cryo-EM map resolved the residues 288 to 294 of the $P_{CT}$ bound to N-RNA. $P_{CT}$ is wedged in between a helix of N (N residues 121-141, helix $\alpha_{I2}$, RSV numbering[22]) on one side and an extended loop of a beta-hairpin of N ($N_{\beta HL}$, N residues 99-112) on the other, burying a total area of ~1100 Å² upon complex formation. We observed a noticeable increase in ordered density of $N_{\beta HL}$ upon P-binding, compared to the apo form (black arrow, Fig. 3B). Of note, $N_{\beta HL}$ is also predicted to be slightly disordered in our Metapredict plots (black arrow in N disorder plot, Fig. 3A). We observe that the P peptide wraps around a central arginine at position R132 (Fig. 3D) belonging to helix $\alpha_{I2}$.P hooks onto this arginine by packing the hydrophobic surfaces of $P_{CT}$ side-chains P-M294, P-L292, P-I289, and P-Y290 against the aliphatic chain of R132, thereby completely surrounding it (Fig. 3D). R132 is additionally stabilized and kept in position through a polar contact with D128. Consistent with this structural data, R132 is highly conserved in *Pneumoviridae* (Supplementary Fig. 6). The C-terminal methionine of P (P-M294) inserts into a hydrophobic cavity on N formed by L46 and M135. Moreover, the face of the aromatic ring of P-Y290 packs against three hydrophobic residues located within $N_{\beta HL}$: L100, I103, and L111 (Fig. 3D). Lastly, viewed in the context of an N-RNA assembly, the $P_{CT}$ binding site is located such that P attaches to a region of the nucleocapsid that protrudes at a far radius from the helical axis (Fig. 3E). This may facilitate accessibility to P-binding, even in light of the conformational heterogeneity of the nucleocapsid.

Past studies have utilized functional and biochemical assays to investigate the HMPV N-RNA/P interaction[31,32]. Our structural observation of the importance of P residues P-M294, P-L292, P-Y290, and P-I289 for the interaction with N-RNA is fully in line with the previous studies, which determined that single substitutions of these residues (compare with Fig. 3D) to alanine completely disrupted the interaction. The same observation could be made with substitutions of N residues R132 and D128, confirming our structure and indicating that D128 is important to orient R132 in a manner that facilitates P-binding[31]. In addition, a MSA incorporating $P_{CT}$ sequences from *metapneumovirus* members shows the conservation of hydrophobic residues in positions 294, 292, 290, and 289 (HMPV numbering), highlighting their importance in clamping onto N (Fig. 3F). Consistently, interacting hydrophobic residues at positions 100 and 111 (HMPV numbering) are also conserved in the N protein, while position 103 is not conserved (Fig. 3G). The $P_{CT}$ sequence of the *orthopneumovirus* RSV diverges from *metapneumovirus* sequences (Fig. 3F), suggesting that the specific interactions between N and P residues may differ between genera.

To investigate the dynamics of $P_{CT}$ bound to N-RNA we carried out all-atom explicit-solvent MD simulations of the triple complex. To circumvent artifacts stemming from the simulation of individual

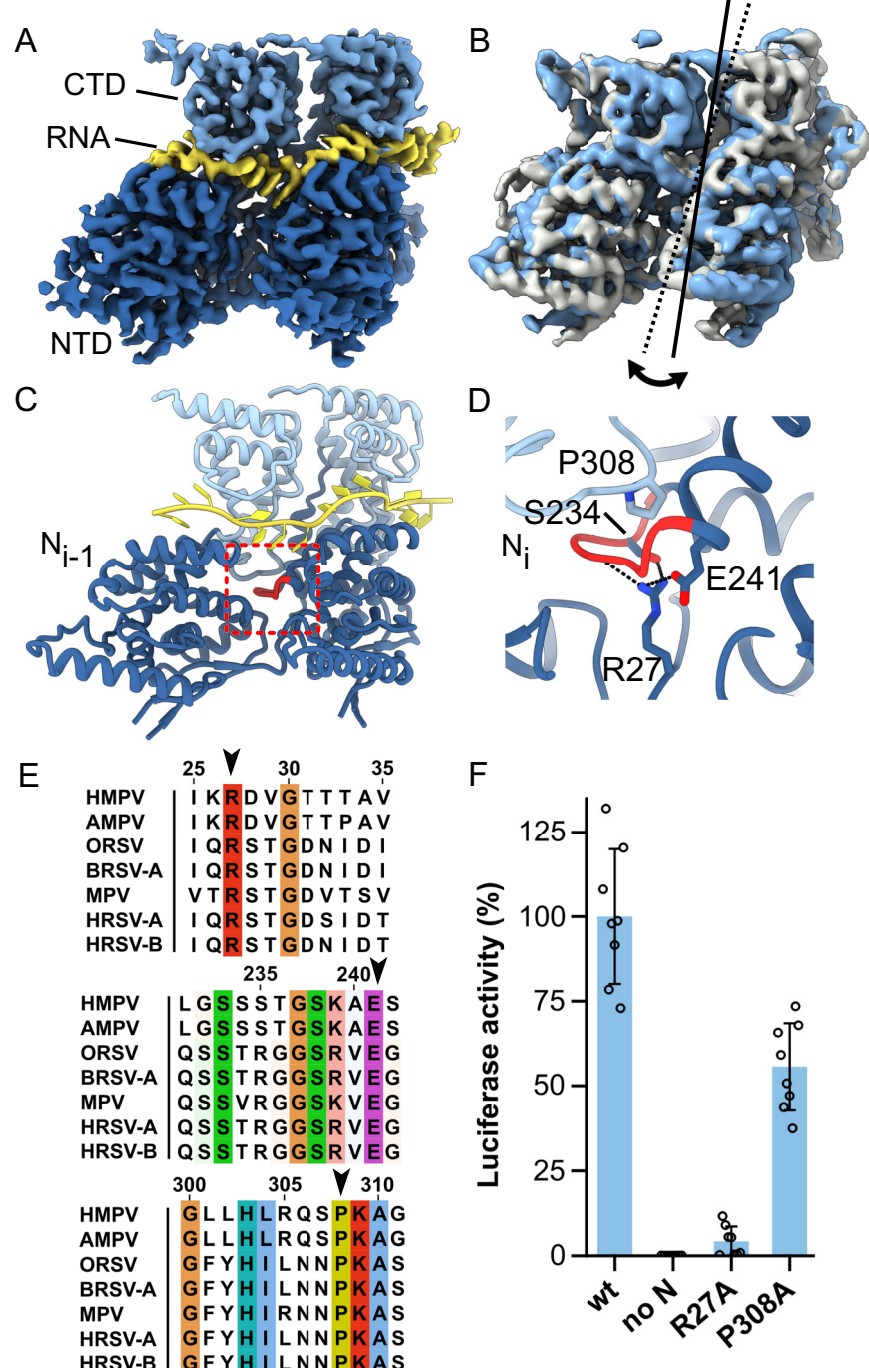

**Fig. 2 | N protomers are hooked together via an inserted loop. A** Cryo-EM map from local refinement. For clarity, only the map surrounding two N-RNA protomers is depicted. Domains of N and the bound RNA are labeled. **B** 3D variability analysis (3DVA) was carried out with the N-RNA dimer. The density maps of starting point and end point of the first variability component are shown in grey and light blue, respectively. A transverse tilting motion could be observed within assembled N-RNA protomers (indicated with continuous and dotted axes). **C** Refined model of the N-RNA dimer. A loop (colored in red) of the $N_i$ protomer is inserted into the neighboring $N_{i-1}$ protomer. **D** Close-up view of the region marked with a red square in panel C. Residues are shown in stick representation. The inserted loop is clamped from both sides via P308 and R27 from $N_{i-1}$. S234 and E241 further stabilize the

interaction through H-bonding. **E** Multiple sequence alignments (MSAs) of N sequences from Pneumoviridae. The conserved positions of R27, E241, and P308 are indicated with arrows. Abbreviations and UniProt accession codes: HMPV (human metapneumovirus, NCAP_HMPVC), AMPV (avian metapneumovirus, NCAP_AMPV1), ORSV (ovine respiratory syncytial virus, NCAP_ORSVW), BRSV-A (bovine RSV type A, NCAP_BRSVA), MPV (murine pneumonia virus, NCAP_MPV15), HRSV-A (human RSV type A, NCAP_HRSVA), HRSV-B (human RSV type B, NCAP_HRSVB). **F** HMPV polymerase activity in the presence of mutations in N relative to strain NL/00/1, assessed by minigenome assay in BSRT7/5 cells. Data points represent the values of quadruplicates from two independent experiments. Data are presented as mean values +/- SD. Source data are provided as a Source Data file.

protomers lacking neighbors, we simulated 5-mers of assembled N-RNA/P. The overall conformation of the N-RNA protomers was consistent throughout the simulations (Fig. 4A). The RNA remained tightly bound, with low root-mean-square-deviations (RMSDs) and low

fluctuations in the number of RNA-protein hydrogen bonds over the simulation times (Fig. 4B). Comparison of simulations with and without $P_{CT}$ indicated that the presence of the peptide slightly increased RMSDs of RNA, however, the number of H-bonds remained consistent

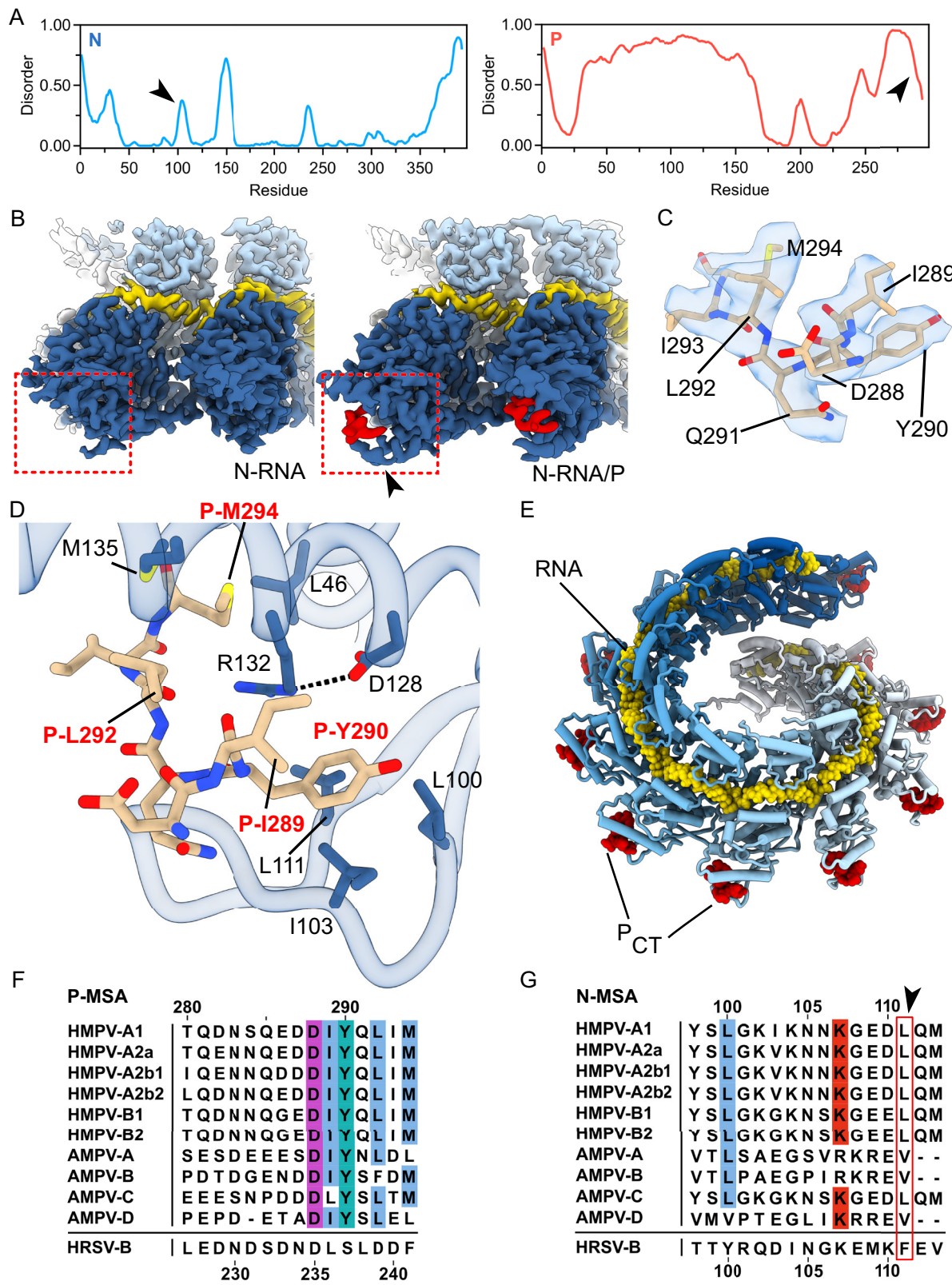

(Fig. 4B). In contrast, substantially higher RMSD values were observed for many of the copies of P-peptide bound to N (Fig. 4C) and the peptide displayed large conformational flexibility (black arrow in Fig. 4A, D). Notably, in one case a peptide completely dissociated from its starting N protomer (black arrow, Fig. 4C), and re-bound at a neighboring copy of N, displacing an already present $P_{CT}$ (Fig. 4E and Supplementary Movie 2). These data indicate that the interaction between $P_{CT}$ and N-RNA is flexible and suggest that P has the ability to

dynamically unbind and rebind on simulated timescales that are notably rapid (i.e. tens to hundreds of nanoseconds).

## Interaction between $N_{\beta HL}$ and $P_{CT}$ is important for polymerase activity and IB formation

Next, we investigated the functional role of the interaction between $P_{CT}$ and the flexible $N_{\beta HL}$ within cells using minigenome assays. We focused our mutations on the residues in positions 100, 103, and 111 of

**Fig. 3 | Cryo-EM structure of the N-RNA/P interface. A** Disorder prediction plots of HMPV N (blue, left) and P (red, right). Arrows, see accompanying text. **B** Comparison of locally refined cryo-EM maps before (left) and after (right) incubation with the $P_{CT}$. The binding region is highlighted with a red square. The density of the $P_{CT}$ is colored in red. Increased ordering of density is observed in N residues 99-112 after the addition of $P_{CT}$ (black arrow). **C** Density of the $P_{CT}$ with refined atomic model. The cryo-EM density is displayed as a transparent surface, the model is shown in stick representation with labeled P residues. **D** Close-up view of the $P_{CT}$ binding site. N residues are shown as blue sticks. P residues are shown as brown sticks with red labels. **E** Model of N-RNA/$P_{CT}$ protomers rigid body fitted into the map of spiral particles incubated with $P_{CT}$. $P_{CT}$ is labeled and depicted as red spheres. **F** MSA of P sequences from different genotypes of human and avian metapneumoviruses (HMPV and AMPV, respectively). The unaligned C-terminal region of the orthopneumovirus HRSV-B is shown below for comparison. GenBank accession codes: HMPV-A1 (AF371337.2), HMPV-A2a (KC403981.1), HMPV-A2b1 (KC562220.1), HMPV-A2b2 (MN745085.1), HMPV-B1 (MW221990.1), HMPV-B2 (MW221994.1), AMPV-A (AY640317.1), AMPV-B (MH745147.1), AMPV-C (KC915036.1), AMPV-D (HG934339.1). **G** MSA of N sequences from metapneumovirus. Hydrophobic residues are conserved at positions 100 and 111 (indicated with red box) of the alignment. Abbreviations and accession codes as in (**F**).

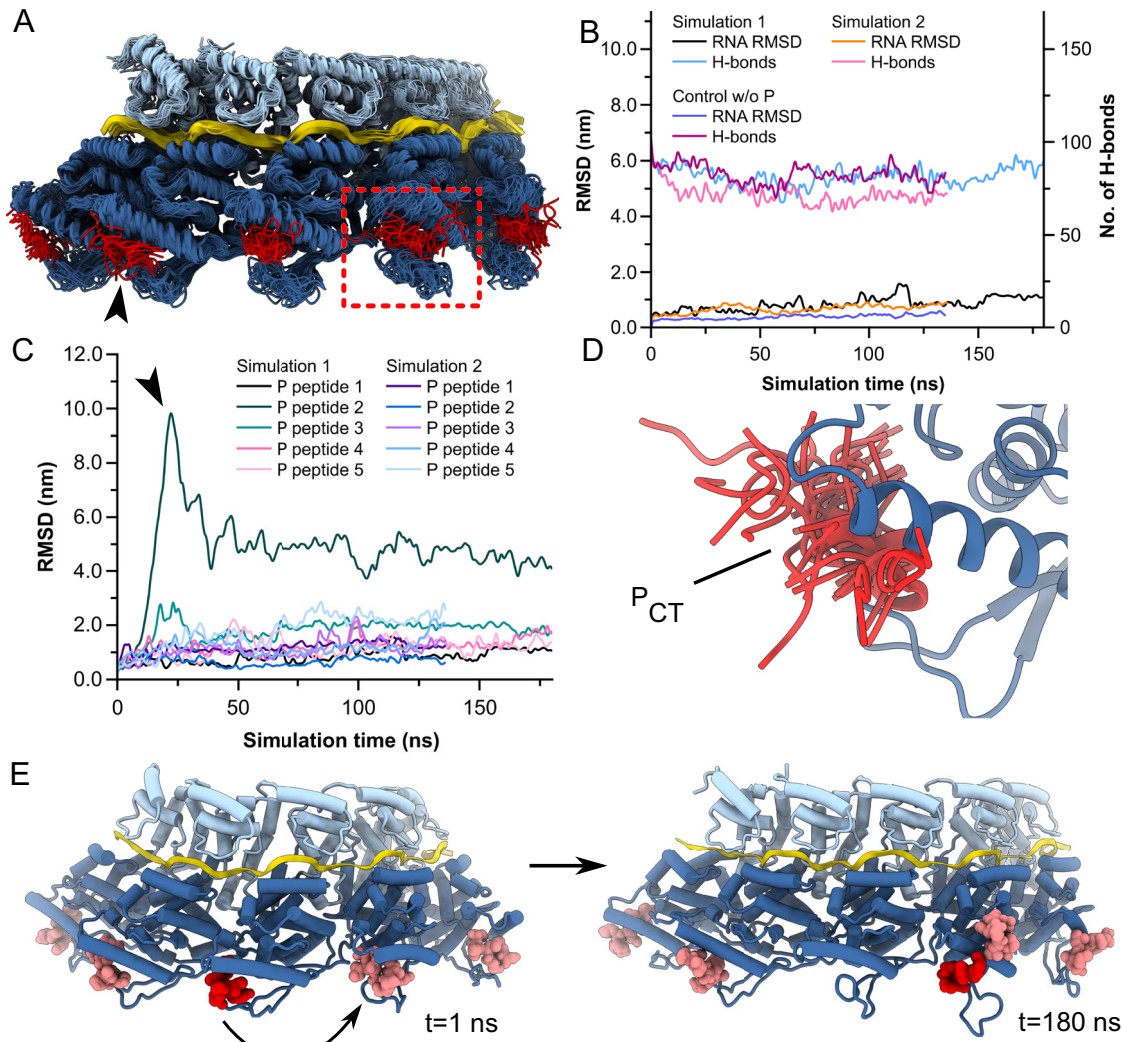

**Fig. 4 | MD simulations of N-RNA/$P_{CT}$. A** Overlayed structural ensemble of a simulated trajectory of a N-RNA/P 5-mer. Coordinates were extracted every 5 ns. $P_{CT}$ was highly dynamic, even in the bound state (black arrow). **B** Time evolution of the number of hydrogen bonds and the root-mean-square-deviations (RMSD) of bound RNA molecules throughout the simulation trajectories. Data are shown for duplicate simulations and a control simulation without $P_{CT}$. **C** Time evolution of RMSDs of $P_{CT}$ peptides throughout the simulation trajectories. High RMSD values were observed for multiple peptides. In one case a peptide unbound and rebound at a neighboring N-RNA protomer (trajectory indicated by black arrow). **D** Ensemble of $P_{CT}$ conformations highlighting its flexibility. For clarity only one ensemble member is shown for N. **E** Snapshots of a simulated trajectory at t = 1 ns (left) and t = 180 ns (right). $P_{CT}$ peptides are colored in shades of red. In the evolution of one trajectory a copy of the $P_{CT}$ (colored in dark red) unbinds, and rebinds at a neighboring N-RNA protomer, displacing an already present $P_{CT}$ (indicated with curved arrow). Source data are provided as a Source Data file.

N, which pack against Y290 of the $P_{CT}$ (see Fig. 3C). Whereas mutations of residue 103 poorly impacted the polymerase activity, consistent with its low degree of conservation (Fig. 3G), mutations of residues L100 and L111 resulted in a noticeable reduction of polymerase activity in the minigenome assay (Fig. 5A). We confirmed by Western blot that this was not due to large variations in expression levels of mutant N proteins (Supplementary Fig. 4). The strongest effect for a single substitution was observed for the leucine at position 111. Exchange to an alanine (L111A) resulted in a polymerase activity decrease to below 50% while an introduction of a negative charge (L111E) led to a decrease to ~10% activity, compared to wild-type. A triple substitution mutant of position 100, 103, and 111 to alanine reduced the activity to around a

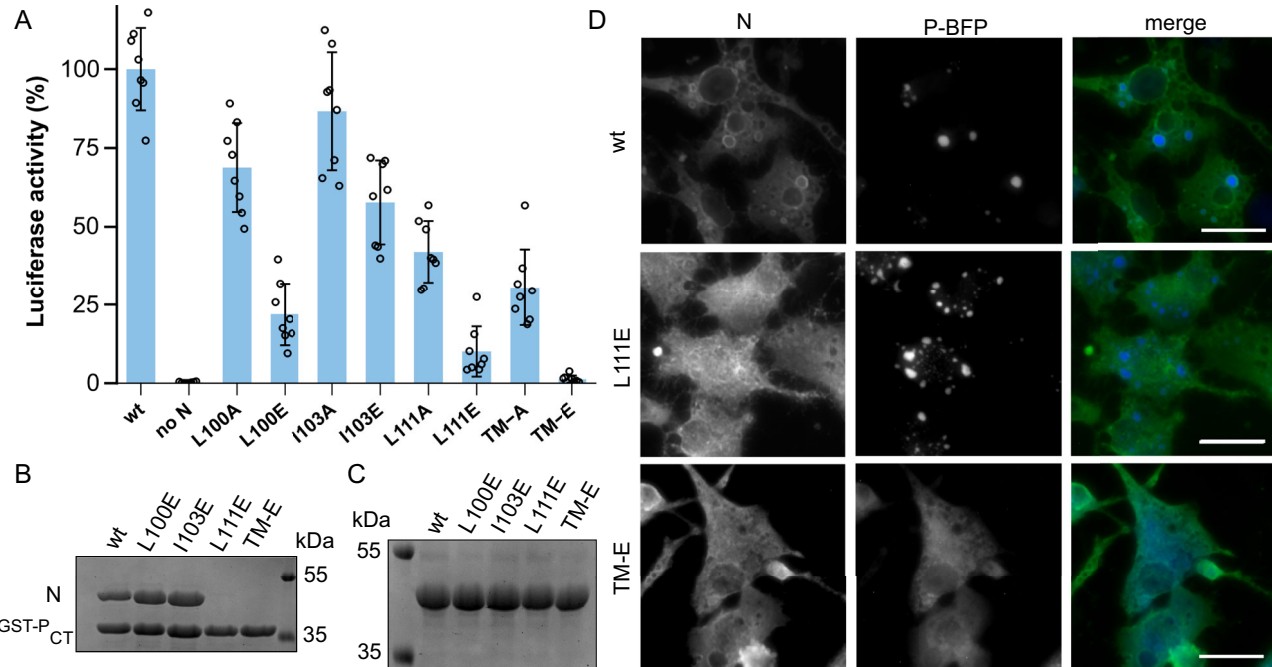

**Fig. 5 | Functional role of the $N_{\beta HL}$ in $P_{CT}$ recognition. A** HMPV polymerase activity in the presence of mutations in N relative to strain NL/00/1, assessed by minigenome assay in BSRT7/5 cells. TM-A denotes a triple mutant with alanines in positions 100, 103, and 111. TM-E denotes a triple mutant with glutamates in positions 100, 103, and 111. Data points represent the values of quadruplicates from two independent experiments. Data are presented as mean values +/- SD. **B** GST-$P_{CT}$ and 6xHis-tagged N proteins (wt and mutants as in A) were co-expressed and co-purified using a GST-tag (one independent repeat of the purification was performed). A SDS-PAGE analysis of the purification products is shown. **C** SDS-PAGE analysis of N (wt and mutants as in A) purified via a 6xHis-tag, demonstrating that these are soluble (one independent repeat of the purification was performed). **D** Fluorescence microscopy of BSRT7/5 cells transfected with plasmids encoding a blue fluorescent protein (BFP)-tagged HMPV P and HMPV N (wt and mutants as in **A**). Cells were fixed 24 h post-transfection. N protein was stained by labeling with rabbit polyclonal anti-N antibody, and nuclei were stained with Hoechst 33342. Scale bars, 20 μm. The experiment was repeated independently twice with similar results. Source data are provided as a Source Data file.

third (~30%), while a corresponding triple mutant with substitutions to glutamic acid in these positions abrogated polymerase activity (Fig. 5A).

Next, we explored the effect of the most impactful substitutions on the ability of N and P to physically interact or to form IBs. We first assessed the ability of a GST fused to $P_{CT}$ to pull down N (wt or mutant) upon co-expression in bacteria (Fig. 5B, uncropped gels in Supplementary Fig. 7). While substitutions of residues 100 and 103 with glutamate retained the association with N, constructs possessing the L111E substitution were not able to copurify N. We also assessed that the loss of interaction was not due to a defect of N expression or solubility, by purifying the individual N proteins expressed in bacteria (Fig. 5C). Although it might be possible that these mutations affect N-RNA assembly, this is rather unlikely as they are located at a flexible, solvent-exposed loop at the periphery of N.

The expression of P and N is sufficient for the formation of pseudo-IBs in cells[28,32]. To test the effect of $N_{\beta HL}$ mutants on IB formation, we co-transfected BSRT7/5 cells with plasmids encoding N (wt and mutants) and BFP-tagged P and imaged the cells by fluorescence microscopy. The P-BFP fluorescence allowed us to observe pseudo-IBs, while labeling with anti-N antibodies allows only the observation of inclusion outlines, likely due to inaccessibility of the interior. Compared to the wt condition, pseudo-IBs formed in the presence of L111E mutant of N were more numerous and smaller, suggesting a defect of nucleation. In line with the pull-down and minigenome assays, the N triple mutant TM-E resulted in a total loss of IB formation, with the P protein being distributed throughout the cytoplasm. Together, these results confirm that $N_{\beta HL}$ is in contact with P, and that perturbation of this contact leads to a loss of polymerase activity and affects LLPS induced by the interaction of N and P proteins.

## Discussion

The viral RNA genome of all nsNSVs is constitutively encapsidated by the nucleoprotein N, forming the nucleocapsid. Packaging within this ribonucleoprotein complex is thought to protect the genome from degradation by cellular RNases but also from the recognition by sensors of the cellular immune system such as RIG-I. However, a consequence of the encapsidation is that it is not naked RNA but rather the N-RNA complex which serves as a template for the viral polymerase L. The P protein is an essential cofactor of L, which interacts with both L and N-RNA, functioning as a flexible tether. The recent cryo-EM structures of *Pneumoviridae* L/P complexes revealed that the central oligomeric domain of the P tetramer and sections of the four C-terminal domains of tetrameric P interact with L[24,61,62]. These structures also showed that each of the C-terminal domains of P adopts specific and different conformations upon interaction with L, highlighting the structural versatility of P which is required for the proper functioning of L/P. Of note, in the HMPV structure the last C-terminal residues of P, identified as essential for the interaction with the nucleocapsid[31,32], were not resolved. The structures imply that one tetramer of P can interact with both L and the nucleocapsid simultaneously. Residues of HMPV P and N involved in the P-nucleocapsid interaction have been identified[31,32]. Previous crystallographical investigations of this N-P interaction in the related pneumovirus RSV utilized a truncated N protein and revealed density for only the last two residues of P (Supplementary Fig. 6)[46], perhaps due to constraints of crystal packing. The results of this study suggested that the C-terminus of P may be flexible upon binding N, which agrees with the weak strength of this interaction (in the micromolar range)[46]. Previous attempts to elucidate how L/P attaches to N-RNA thus remain limited, and the impact of P binding on full-length N is still unknown.

Here, using cryo-EM and full-length N, we were able to determine the structures of assembled and RNA-bound HMPV N in different oligomeric states, including a spiral particle reminiscent of a helical turn, and with improved resolution compared to a previous crystal structure[33]. Analysis of our structural data revealed heterogeneity in the relative geometries of assembled N protomers, supporting the notion of a highly flexible nucleocapsid. Furthermore, we identified a key N-N interaction involving the loop ranging from residues 232-239 of the $N_i$ protomer which inserts into the neighboring $N_{i-1}$ protomer. We found that residue R27 of the loop was sensitive to mutation and a single exchange to alanine had a major impact on polymerase activity. However, further structural characterization of N-RNA oligomers of R27 mutants are necessary to determine the precise role of this interaction for nucleocapsids.

We were successful in obtaining the first structural data on the HMPV $P_{CT}$-N-RNA interaction. Utilizing an approach that made use of a large molar excess of $P_{CT}$ vs. N, computational symmetry expansion, and local refinement we were able to resolve $P_{CT}$ bound to the NTD of assembled and RNA-bound N. Our structure reveals how P binds by hooking itself around a centrally positioned arginine of the NTD, thereby latching on to N-RNA. The $P_{CT}$ interacts with the partially disordered $N_{\beta HL}$ region of the NTD, which becomes more ordered upon binding. Furthermore, in the apo-state the $P_{CT}$ itself has been characterized as having high disorder propensity[34,35]. Such folding upon binding events of intrinsically disordered regions are thought to facilitate highly specific interactions, yet possess low affinities and rapid dissociation kinetics[63]. We recognize that our experimental conditions involving ~100-fold molar excess of $P_{CT}$ would not be the case in vivo, but likely assisted in shifting the equilibrium to a bound state, enabling resolving the site.

To explore the dynamics of the $P_{CT}$/N-RNA complex we turned to all-atom MD simulations of assembled 5-mers. We observed that the presence of the peptide did not have a major impact on the stability of RNA bound to the nucleoprotein cleft. This suggests that $P_{CT}$ attachment alone is not sufficient for inducing the release of RNA from the nucleocapsid – a fundamental requirement to enable L/P access its template. Perhaps it is instead the free CTD-arm of N that is involved in dislodging RNA at the 3′-end of nucleocapsids and thereby assists L/P binding the initial nucleotides. In line with this notion, the CTD-arm has previously been shown to possess a wide spectrum of possible conformations[23] and affinity for the RNA-binding cleft of N[33]. In addition, the simulations revealed substantial conformational flexibility of $P_{CT}$ within the N-RNA/P complex. In one instance we observed the dissociation and rebinding of one of the simulated peptides. This is in line with the disordered character of P, but also with previous reports in RSV characterizing the interaction as giving rise to a fuzzy complex[59,64].

In the cryo-EM structure of HMPV L/P, the P tetramer binds L in an asymmetrical manner: the C-termini of 3 protomers of P are bundled together on one side of the polymerase complex and are resolved to around residue 236, while another P protomer on the opposite side is resolved up to 266 of a total of 294 residues and winds around the nucleotide entry channel[24]. It was previously suggested that the three bundled P protomers interact with N-RNA, while the P protomer at the nucleotide entry channel is more directly involved in polymerase function[24]. This notion is fully consistent with the fact that in a structure of the related RSV L/P the $P_{CT}$ of the protomer at the nucleotide channel is found to fully attach to L and is thus unavailable for N-RNA binding[62]. The observation that the 3 remaining P protomers from HMPV only interact with L up to residue 236 implies a long linker to the N-RNA binding site (~50 residues) potentially able to span large distances between L/P and N-RNA. In addition, this linker region is potentially highly flexible[35], suggesting that the N-RNA binding sites of HMPV P can sample large capture radii, and L/P may be able to bind to N-RNA over considerable distances or over multiple turns of the helical

nucleocapsid. However, structural data of nucleocapsid-bound L/P is required to substantiate these ideas.

Taking together the properties of the N-RNA/P interaction, we suggest that its evolution is constrained by a need to balance sufficient tethering of L to the nucleocapsid with being able to unbind/rebind with low energy barriers during transcription. RNA-polymerases exert pulling forces in the tens of Piconewtons range[65] and in the case of L/P this needs to be sufficient to dislodge tetrameric P from any given position on the nucleocapsid while it is being traversed by L/P. In the hypothetical case of a $P_{CT}$ binding too tightly to N-RNA, the pulling force might not be able to overcome this tethering, leading to a stalled complex. At the same time, the interaction needs to be strong enough to recruit L/P for transcription. Several decades ago a cartwheeling model was proposed for the Sendai paramyxovirus[66], suggesting that "P is envisaged to "walk" on the template via the simultaneous breaking and reforming of subunit-template contacts". Our data fully support the applicability of this classic model for the *Pneumoviridae* L/P complex, in which it scuttles along the nucleocapsid template during processive RNA synthesis using the C-terminal domains of tetrameric P as legs.

Our study provides a structural blueprint for a potentially druggable interface in HMPV. Previous attempts have been made to target the RSV N/RNA-P interaction, resulting in the compound M76, which inserts into the $P_{CT}$ binding site on RSV N[46]. Unfortunately, antiviral inhibition through M76 was limited. More recently, the 1,4-benzodiazepine-derived compound EDP-938 shows promising antiviral activity against RSV-A and RSV-B, a relatively high barrier against resistance, efficacy in non-human primates, and was superior to placebo in a randomized, double-blind challenge trial[67,68]. As of 2023, EDP-938 is still undergoing phase II clinical trials. Based on the mapping of resistance mutations, it was concluded that the compound targets the RSV N protein. Interestingly, the by-far strongest resistance mutant to EDP-938 (M109K, responsible for a 67-fold worsening of the $EC_{50}$) is located in a region corresponding to the $N_{\beta HL}$ of HMPV (red spheres in Fig. 6A). Comparison with our N-RNA/P complex (Fig. 6A, B), with its extended interaction surfaces, suggest that M109 may be involved in transiently clamping down on the $P_{CT}$ in RSV, by analogy to HMPV. This is also in-line with our functional data which show that $N_{\beta HL}$ mutations modulate the interaction with the peptide. These observations strengthen the hypothesis that the N-RNA/P interaction may be the target of EDP-938. However, although EDP-938 treatment was shown to reduce the accumulation of viral RNA early in the infection, the matter is complicated by the fact that EDP-938 had no effect on transcription in a minigenome assay[67]. Whether this may be due to experimental constraints or because there are additional unknown factors in the mechanism of the compound will need to be explored in future work.

## Methods

### Plasmids for pulldowns and minigenome assays

The published plasmids[31] pET-N and pGEX-$P_{CT}$ encoding for N with a C-terminal 6xHis tag and for the C-terminal residues 200-294 of P fused to GST respectively, with N and P sequences derived from HMPV strain CAN97-83, were utilized. Moreover, the published HMPV minigenome plasmid containing Gaussia/Firefly luciferases and the plasmids pP, pN, pL, and pM2-1 corresponding to the sequences of HMPV strain NL99-1 cloned into the pCite vector, as well as pP-BFP vector were also utilized[31]. Of note, position 103 of the N sequences of strains CAN97-83 (valine) and NL99-1 (glycine) differ from the reference strain NL/00/1 used for the structural studies (isoleucine). Point mutations were introduced in the N sequence by site-directed mutagenesis using the Q5 site-directed mutagenesis Kit (New England Biolabs). Primers used for mutagenesis are shown in Supplemental Table 2. Sequence analysis was carried out to check the integrity of all constructs.

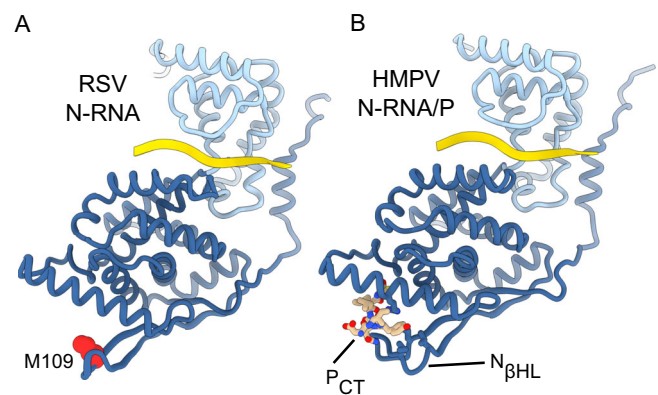

**Fig. 6 | Comparative analysis of N-M109 resistance mutation against EDP-938 in RSV. A** Structure of a RSV N-RNA protomer (PDB ID: 2WJ8). RNA is depicted as yellow ribbon. The residue M109 is indicated and depicted as red spheres. **B** Structure of a HMPV N-RNA/P protomer. The $P_{CT}$ is indicated and depicted as sticks.

## Antibodies

Antisera used in this study included polyclonal rabbit antisera raised against recombinant HMPV N expressed in bacteria. A mouse monoclonal anti-tubulin (Sigma, product number T6199, clone DM1A, lot 115M4796V) and horse-radish peroxidase-coupled secondary antibodies directed against mouse IgG (SeraCare, material number 5450-0011, lot 10430730) and rabbit IgG (SeraCare, material number 5450-0010, lot 10437708) were also used for immunoblotting. A secondary antibody directed against rabbit IgG coupled to Alexafluor-488 (Invitrogen, Catalog #A-11059, lot 682615) was used for immunofluorescence experiments.

## Cell culture and transfections

BHK-21 cells (clone BSRT7/5) constitutively expressing the T7 RNA polymerase[69] were grown in Dulbeco Modified Essential Medium (Lonza) supplemented with 10% fetal calf serum (FCS), 2 mM glutamine, and antibiotics. The cells were grown at 37 °C in 5% $CO_2$ and transfected using Lipofectamine 2000 (Invitrogen) as described by the manufacturer.

## Minigenome replication assay

BSRT7/5 cells at 90% confluence in 96-well dishes were transfected using Lipofectamine 2000 (Invitrogen) with a plasmid mixture containing 62.5 ng of pGaussia/Firefly minigenome, 62.5 ng of pN, 62.5 ng of pP (WT and mutants), 31.25 ng of pL, 31.25 ng of pM2-1, as well as 15.6 ng of pSV-β-Gal (Promega) to normalize transfection efficiencies. Cells were harvested at 24 h post-transfection and lysed in 100 µl of Firefly lysis buffer (30 mM Tris [pH 7.9], 10 mM MgCl2, 1 mM dithiothreitol [DTT], 1% [vol/vol] Triton X-100, and 15% [vol/vol] glycerol). Then 50 µL of lysates were mixed to 50 µl D-luciferine (Luciferase assay system, Promega) and the Firefly luciferase activity was determined for each cell lysate with an Infinite 200 Pro (Tecan, Männedorf, Switzerland), and normalized based on β-galactosidase (β-Gal) expression. Transfections were done in quadruplicate, and each independent transfection experiment was performed up to four times. For protein expression analysis, cells were lysed in Laemmli buffer, boiled, and analyzed by Western blotting (WB) using the following primary antibodies: rabbit polyclonal raised in-house against recombinant HMPV N (1:1000 dilution) and mouse monoclonal against tubulin (Sigma, 1:1000 dilution, product number T6199, clone DM1A, lot 115M4796V). The following secondary antibodies were utilized: anti-mouse coupled to horse-radish peroxidase (SeraCare, 1:10000 dilution, material number 5450-0011, lot 10430730) and anti-rabbit coupled to horse-radish peroxidase (SeraCare, 1:10000 dilution, material number 5450-0010, lot 10437708). Uncropped Western blots are shown in Supplementary Fig. 4.

## Fluorescence microscopy

Immunofluorescence microscopy was performed with cells grown on coverslips and previously transfected with pP-BFP and pN (WT or mutants), at a ratio of 1:1. At 24 h post-transfection, cells were fixed with 4% paraformaldehyde (PFA) for 20 min, made permeable and blocked for 30 min with PBS containing 0.1% Triton X-100 and 3% bovine serum albumin (BSA). Cells were then successively incubated for 1 h at room temperature with primary and secondary antibody mixtures diluted in PBS containing 3% BSA and 0.05% Tween20. Utilized antibodies: rabbit polyclonal raised in-house against recombinant HMPV N (1:5000 dilution) and secondary anti-rabbit IgG coupled to Alexafluor-488 (Invitrogen, 1:2000 dilution, catalog #A-11059, lot 682615). Coverslips were mounted with ProLong Gold antifade reagent (Invitrogen) and observed with an inverted fluorescence microscope (Zeiss Axiovision). Images were analyzed with ImageJ 1.53.

## Expression and purification of recombinant proteins

*E. coli* BL21 bacteria (DE3) (Novagen) transformed with pET-N (WT or mutants) alone or with pGEX-$P_{CT}$ were grown at 37 °C for 8 hours in 100 ml of Luria Bertani (LB) medium containing 100 µg/ml ampicillin or ampicillin (100 µg/ml) and kanamycin (50 µg/ml), respectively. The same volume of LB was then added and protein expression was induced by adding 80 µg/ml isopropyl-ß-D-thio-galactoside (IPTG) to the medium. The bacteria were incubated for 15 hours at 28 °C and then harvested by centrifugation. For purification using the GST-tag, bacteria were re-suspended in lysis buffer (50 mM Tris-HCl pH 7.8, 60 mM NaCl, 1 mM EDTA, 2 mM DTT, 0.2% Triton X-100, 1 mg/ml lysozyme) supplemented with complete protease inhibitor cocktail (Roche), incubated for 1 hour on ice, sonicated, and centrifuged at 4 °C for 30 min at 10,000 g. Glutathione-Sepharose 4B beads (GE Healthcare) were added to clarified supernatants and incubated at 4 °C for 3 hours. Beads were then washed with 10 volumes of lysis buffer, then three times with PBS 1X, and then stored at 4 °C in an equal volume of PBS.

For purification of N-6xHis fusion protein purification, bacterial pellets were re-suspended in lysis buffer (20 mM Tris-HCl pH8, 500 mM NaCl, 0.1% TritonX-100, 10 mM imidazole, 1 mg/ml lysozyme) supplemented with complete protease inhibitor cocktail (Roche). After sonication and centrifugation, lysates were incubated 30 min with chelating Sepharose Fast Flow beads charged with $Ni^{2+}$ (GE Healthcare). Beads were then successively washed in the washing buffer (20 mM Tris-HCl, pH 8, 500 mM NaCl) containing increasing concentrations of imidazole (25, 50, and 100 mM), and proteins were eluted in the same buffer with 500 mM imidazole. For SDS-PAGE analysis, samples were prepared in Laemmli buffer, denatured for 5 min at 95 °C, and separated on 12% polyacrylamide gel colored by Coomassie brilliant blue.

## Expression and purification of HMPV N-RNA for structural analysis

The full-length N protein gene from human metapneumovirus strain NL/00/1 (GenBank: YP_009513265.1) was cloned into the pOPINE bacterial expression vector, also coding for a C-terminal His-tag, using the In-Fusion system (Takara Clontech, Mountain View, CA)[70]. The vector DNA sequence was verified by means of Sanger sequencing. Rosetta2 *E. coli* cells containing the plasmid were grown at 37 °C in terrific broth (TB) containing ampicillin (100 µg/ml). Once an $OD_{600}$ of 0.8 was reached, the cells were cooled to 18 °C and then induced by the addition of isopropyl β-D-1-thiogalactopyranoside (IPTG) to a final concentration of 1 mM. Cells were harvested after overnight growth (18 °C, 20 min, 4000 x g). Cell pellets were resuspended in 40 mL of wash buffer (25 mM Tris-HCl, pH 8.0, 1 M NaCl) per litre of bacterial culture, and lysed by means of sonication. The resultant lysate was clarified (4 °C, 45 min, 50000 x g) and the supernatant was filtered and loaded on a column containing Ni²-NTA (nitrilotriacetic) agarose

(Qiagen, Netherlands). The column was equilibrated with wash buffer and the bound protein was washed. The protein was eluted in elution buffer (25 mM Tris-HCl, pH 8.0, 1 M NaCl, 300 mM Imidazole), and the resultant eluate was further purified by size exclusion chromatography, using a Superose6 10/300 column (GE Healthcare, United Kingdom), equilibrated in SEC buffer (25 mM Tris-HCl, pH 8.0, 1 M NaCl). The protein was then buffer exchanged into 25 mM Tris-HCl, pH 8.0, and 250 mM NaCl using a PD10 column (GE Healthcare).

### Preparation of N-RNA/P complexes
A synthetic peptide of P covering the sequence N-EDDIYQLIM-C was obtained from GenScript. The peptide was dissolved in deionised water and dosed with a drop of 5 M $NH_4OH$ to improve solubility. N-RNA preparations were mixed with the resulting peptide solution in a ~100-fold molar excess of P peptide to N. The N-RNA and $P_{CT}$ mixture was incubated at 4 °C overnight.

### Cryo-EM data collection and processing
Cryo-EM experiments were carried out at the Oxford Particle Imaging Centre (OPIC). 3 μL of N-RNA (1.0 mg/ml) or N-RNA/$P_{CT}$ (0.5 mg/ml) sample was placed onto glow-discharged Quantifoil R2/1 Cu300 holey carbon grids (1 μm spacing, 2 μm holes, copper mesh) (Quantifoil, Germany), blotted for 3.5 s before flash-freezing in liquid ethane using a Vitrobot mark IV (FEI) at 95 – 100 % humidity. Cryo-EM data were collected using SerialEM 3.6 on a 300 kV Titan Krios microscope (Thermo Fisher Scientific) fitted with a K2 Summit (Gatan) direct electron detector and GIF Quantum energy filter. Data collection parameters are listed in Supplementary Table 1 for both datasets. Motion correction of cryo-EM movies and contrast transfer function (CTF) parameters were estimated on the fly using cryoSPARC live 2.15[71].

All single-particle reconstructions were carried out in cryoSPARC 2.1. Particle picking was carried out using a combination of the cryoSPARC blob picker and template picker. The 2D classification was utilized to separate particle subsets of different oligomeric states and remove bad particles. Particles were further cleaned through heterogeneous refinement. The final maps were obtained from the non-uniform refinement job type. Symmetry was imposed only when it was clearly visible in template-free 2D class averages, as was the case for the 10-mer and 11-mer classes (C10 and C11 symmetry imposed, respectively). No symmetry was imposed for the refinement of spiral particles. To improve the density of N-RNA protomers we carried out 5-fold symmetry expansion of the 10-mer particles, followed by local refinement of a N-RNA dimer. Mask creation for the local refinement was based on a N-RNA dimer from PDB ID: 5FVC. We also attempted an approach with 10-fold symmetry expansion and local refinement of a N-RNA monomer, however this yielded worse maps, presumably because a single monomer is not of sufficient size for local refinement (less than 50 kDa). Total particle numbers and numbers of particles used for the reconstruction of every map can be seen in Supplementary Figs. 1 and 5 for N-RNA and N-RNA/$P_{CT}$, respectively, and in Supplementary Table 1. Local resolution maps for all reconstructions are shown in Supplementary Fig. 8.

### Model building and refinement
The model of HMPV N obtained from X-ray crystallography was used as a starting point for further refinement (PDB ID: 5FVC). Iterative cycles of manual refinement and building in COOT 0.9.7[72] and real-space refinement in PHENIX 1.18[73] were used to optimize geometry and fit the density. PHENIX validation tools were used to monitor geometry and density fit parameters. For maps with imposed symmetry (10-mer and 11-mer maps), we refined a single protomer copy and generated symmetry mates for figure preparation. The N-RNA models (from maps without P) encompassed residues 3 to 365 of the N protein, with a gap from residues 100 to 111 where the density was not sufficient for

building a model. Moreover, we observed only very incomplete and diffuse density after residue 365 (the C-arm) and thus refrained from continuing to build the model after this point. For the models of the spiral particles we limited ourselves to rigid body fitting of N-RNA protomers (from the high-resolution refinements) with flexible N-and C-termini truncated, as the spiral particle maps were only of intermediate quality, presumably due to low particle numbers and heterogeneity. For the N-RNA/$P_{CT}$ complex, the local refined map showed much-improved density in the region of residues 100-111 and thus allowed building a continuous model up to residue 365. The density for the P-peptide had the best quality in the N-RNA/$P_{CT}$ local refinement, so this was used to model P residues 288 to 294. The N-RNA/$P_{CT}$ model from the locally refined dimer maps was then used as a basis for the model refinement of N-RNA/$P_{CT}$ 10-mers and 11-mers. For the spiral map we again only refined the overall positions of N-RNA/$P_{CT}$ protomers with truncated termini via rigid-body, due to limited map quality. Model refinement statistics are given in Supplementary Table 1.

### Molecular dynamics simulations
Classical MD simulations were used to study the dynamics of the N-RNA and N-RNA/$P_{CT}$ complexes. A 5-mer of N-RNA complex, with or without the 5 bound $P_{CT}$ peptides, was extracted from the N-RNA/$P_{CT}$ cryo-EM structure. The MD systems were set up using the solution builder tool from CHARMM-GUI input generator[74]. Each system was solvated in a rectangular periodic box which size was determined by the biomolecular extent, resulting in the addition of approximately 200k TIP3P water molecules. The systems were neutralized by adding 150 mM NaCl. The two systems were then energy minimized, equilibrated in NVT ensemble, and simulated for ~180 ns in 2 independent trajectories in GROMACS2021[75] by making use of the CHARMM-GUI webserver provided scripts[76]. The CHARMM36m force field was used for all simulations[77]. The trajectories were analyzed using GROMACS tools to extract RMSDs and a number of hydrogen bonds as a function of simulation time.

### Structure analysis and figure preparation
Structures were inspected and analyzed using tools within UCSF Chimera 1.14 and ChimeraX 1.1[78,79]. Structure figures and movies were prepared in ChimeraX. Multiple sequence alignments were prepared using the PROMALS3D webserver and Jalview 2.11.2.0[80,81], using the Clustal X color scheme. Structure interfaces were analyzed using the COCOMAPS server[82].

### Reporting summary
Further information on research design is available in the Nature Portfolio Reporting Summary linked to this article.

## Data availability
The coordinates and density maps generated in this study have been deposited in the Protein Data Bank (PDB) and the Electron Microscopy Databank (EMDB) and under accession codes: HMPV N-RNA 10mer (PDB ID: 8PDL, EMD-17613), HMPV N-RNA 11mer (PDB ID: 8PDM, EMD-17614), HMPV N-RNA spiral (PDB ID: 8PDN, EMD-17615), local refinement of a HMPV N-RNA dimer (PDB ID: 8PDO, EMD-17616), HMPV N-RNA/P 10mer (PDB ID: 8PDP, EMD-17617), HMPV N-RNA/P 11mer (PDB ID: 8PDQ, EMD-17618), HMPV N-RNA/P spiral (PDB ID: 8PDR, EMD-17619), local refinement of a HMPV N-RNA/P dimer (PDB ID: 8PDS, EMD-17620). The data underlying Figs. 1A, 2F, 4B, 4C, and 5A are provided in the Source Data File. Source data are provided with this paper.

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

## Acknowledgements

We are grateful to B. van den Hoogen (Erasmus MC, Rotterdam) for providing plasmids of the HMPV NL99-1 strain. JDW is supported by a Wellcome Trust DPhil scholarship (218482/Z/19/Z). Electron microscopy was conducted at the OPIC electron microscopy facility, which was funded by a Wellcome JIF award (060208/Z/00/Z) and was supported by a Wellcome equipment grant (093305/Z/10/Z). The Wellcome Trust is also acknowledged for providing administrative support (Grant 075491/Z/04). The work also received financial support from the French Agence Nationale de la Recherche, specific programs ANR DecRisP n° ANR-19-CE11-0017. Computation used the Oxford Biomedical Research Computing (BMRC) facility, a joint development between the Wellcome Centre for Human Genetics and the Big Data Institute, supported by Health Data Research UK and the NIHR Oxford Biomedical Research Centre. Financial support was provided by a Wellcome Trust Core Award (203141/Z/16/Z). The views expressed are those of the author(s) and not necessarily those of the NHS, the NIHR, or the Department of Health.

## Author contributions

J.D.W., C.L., J.F.E., M.G., and M.R. conceived and designed the study. J.D.W. and M.R. prepared the samples for structural analysis. J.D.W., L.C., and M.R. collected cryo-EM data, processed the data, and refined the structural models. H.D., J.F., J.F.E., and M.G. performed functional assays and analyzed data. C.L. and M.R. prepared and performed MD simulations and analyzed the trajectories. J.D.W., M.G., and M.R. wrote the paper with input from all the co-authors. M.G. and M.R. supervised the work.

## Funding

## Competing interests

The authors declare no competing interests.
