## [Peer Review File · Nature Communications]

Structure of the N-RNA/P interface indicates mode of L/P recruitment to the nucleocapsid of human metapneumovirusREVIEWER COMMENTS

Reviewer #1 (Remarks to the Author):

This manuscript reports two structure studies, the N-RNA ring of HMPV and its complex with the C-terminus of phosphoprotein, by cryo-EM. The outcomes are very similar to those of other nonsegmented negative strand RNA (NNSR) viruses, including a closely related virus, RSV. Very little new information is revealed even though the results are confirmative of previous reports.

The structure of N-RNA rings conformed that N protomers may adopt slightly altered interactions between them in the nucleocapsid, depending on the conformation of the nucleocapsid. Such variations are extensively reported previously, including the recent publications as the following:

Nat Commun. 2022 Oct 10;13(1):5980. doi: 10.1038/s41467-022-33664-4

Nat Commun. 2022 Aug 15;13(1):4802. doi: 10.1038/s41467-022-32223-1

which were omitted by the authors. The authors discussed the selected protomer conformation observed in their cryoEM structures of N-RNA rings with a small number of N protomers, which are just a few sampling points. The short spiral N-RNA assembly is essentially the same as others.

For the complex between the N-RNA ring and the C-terminus of P, the structure is the same as previously reported for RSV. It is likely that other residues in the C-terminal region of P are also involved in the interactions with the N-RNA complex. As previously reported for other NNSR viruses, the C-terminal region of P selectively binds the nucleocapsid by interacting with two neighboring N protomers that are only present in the nucleocapsid assembly, although there is a major interaction area with one N protomer. The function of N binding by the C-terminus of P has been reported by this group in 2022, including minigenome assays, and IB formation, with introduction of mutations in P and N. The selection of mutated residues was slightly different in the 2022 publication, which is indicative of conformation-dependent interactions of P C-terminus with N. This proves that mutations selected in this paper are not more informative than those in the 2022 publication.

Overall, the results included in this manuscript hardly add any new knowledge in this area, maybe just some small incremental pieces of new data.

Other concerns are as the following:

Abstract: The authors' statement "To define how L/P recognizes N-encapsidated RNA (N-RNA) we employed cryogenic electron microscopy" is not accurate. The results in this manuscript only defined the interactions between P and N-RNA. Interactions of L/P complex with N-RNA shall include interactions between P and N-RNA, and between L and N-RNA. The authors can only claim that they were to define the interactions between P and N-RNA.

"The nucleoprotein N is a promiscuous RNA-binding protein" is not a correct statement. N itself does not bind RNA per se. RNA is captured only if N can assemble into oligomers (see J Virol. 2008 Jan;82(2):674-82. doi: 10.1128/JVI.00935-07).

"with the remainder being disordered." is not an accurate statement. P protein domains may assume different conformational states so they are not visible in the cryo-EM densities, but that does not mean that these domains are disordered.

"1) it tethers the L/P complex to the nucleocapsid while traversing the genome during transcription/replication," is not completely accurate. Is "it" P? P alone cannot traverse the genome.

"the structural mechanism of P attachment to N-RNA which retains substantial conformational flexibility in the bound state." What is "which"? P or N-RNA?

"HMPV N-RNA rings were expressed" is a sloppy statement. recombinant N was expressed and N assembles N-RNA rings by encapsidating host RNA.

"Mutagenesis of P308 to alanine resulted in a notable decrease of polymerase activity, while the R27A mutation nearly abrogated polymerase activity (Fig. 2F), suggesting that the N-N interaction via the 232-239 loop is important for successful N-RNA assembly." This conclusion is not sound. The mutations have effects on minigenome assay, but there is no data on N-RNA assembly (see J Virol. 2008 Jan;82(2):674-82. doi: 10.1128/JVI.00935-07). The mutations could just cause conformational changes in N to interfere with interactions with L/P complex.

"the C-terminal region of P (PCT)", "the 9 last residues of PCT (N-EDDIYQLIM-C)". The last 9 residues are not "the C-terminal region". It is just the C-terminus.

"These data indicate that the interaction between PCT and N-RNA is flexible and suggest that P has the ability to dynamically unbind and rebind on simulated timescales that are notably rapid (i.e. tens to hundreds of nanoseconds)." This is an overreaching statement. The MD simulation is only on 9 residues of P. It is most likely that there are more extensive interactions between the C-terminal region of P and N-RNA.

"was not due to a defect of N expression or solubility, by purifying the individual N proteins expressed in bacteria (Fig. 5C)" But they did not check if these mutations disrupt proper N-RNA assembly.

"Residues of HMPV P and N involved in the P-nucleocapsid interaction have been identified". Some residues, not all.

"The observed dynamics of N protomer orientations, even in the assembled state, may constitute a prerequisite for conformational changes facilitating access to the RNA genome during transcription by L/P". Overreaching conclusion.

Reviewer #2 (Remarks to the Author):

HMPV and RSV are major human viral pathogens causing serious respiratory infections. While a vaccine has been recently marketed for RSV prevention (<https://www.fda.gov/news-events/press-announcements/fda-approves-first-respiratory-syncytial-virus-rsv-vaccine>), no vaccine exists for HMPV. A better understanding of HMPV replication could help the design of antiviral therapies. Experimental structures are available for RSV and HMPV polymerase L/P complexes as well as for their ring-forming nucleocapsids. However, one key information lacking is how the CTD of the P phosphoprotein interacts with the nucleocapsid that wraps viral genomic RNA. To fill this gap, here, the authors report cryo-EM structures of recombinant HMPV N-RNA bound to PCT (N-RNA/P). They identify a conserved groove at the periphery of the N multimer to which P binds. This interface region is located between a helix and an extended loop of a beta-hairpin of N (N β HL). The authors use a minigenome experiments to validate the interaction as well as MD simulations of a complex between N-RNA/P to study the conformational dynamics of PCT. Overall, this is an important well-executed study. I only have a few comments.

Major issues

1-P4: "The nucleoprotein N is a promiscuous RNA-binding protein with globular N-terminal and C-terminal domains (NTD and CTD, respectively)^{18,29}."

The introduction should spell out more clearly what has been published in terms of N-P interactions eg: ref 29 determined crystal structures of PNTD with recombinant N0 (RNA free N protein). This is to highlight the really novel results reported in this ms for the lay reader.

2- Fig. 3F

HMPV-P: 288-DIYQLIM-294

Is this region truncated in RSV-P ?? One would expect conservation of this functionally important region between RSV-P and HMPV-P ? Please comment on this. Nonetheless, the authors rightly note that the present work is in line with previous mapping of N-P interaction for HMPV reported in refs 27 and 28.

27. Thompson, R. E., Edmonds, K. & Dutch, R. E. Specific Residues in the C-Terminal Domain of the Human Metapneumovirus Phosphoprotein Are Indispensable for Formation of Viral Replication Centers and Regulation of the Function of the Viral Polymerase Complex. *J Virol* e0003023 (2023) doi:10.1128/jvi.00030-23.

28. Decool, H. et al. Characterization of the Interaction Domains between the Phosphoprotein and the Nucleoprotein of Human Metapneumovirus. *J Virol* 96, (2022).

3- One would expect the interaction between PCTD and N to destabilize the interaction between N and RNA to allow threading of RNA into the polymerase for replication/transcription. This does not seem to be supported by the manuscript. This should be discussed in terms of a revised model of replication/transcription. Maybe using a figure or a video capturing the key results of the present work.

4- R132 from N-HPMV appears to play a key role for the interaction with PCT, how evolutionary conserved is R132?

5- Fig. 1 A: Elution profile from superose 6. Some MW standards should be added to this panel for consistency with the other panels.

6- The two supplementary videos are not very informative. Is it possible to improve them in terms of content (eg adding labels, arrows, captions). This is an important work that should be well presented and informative videos could help.

Minor issues

1-P3 Currently, there are no licensed vaccines or specific therapeutics for the treatment of HMPV infections.

There is now a licensed vaccine for the closely related RSV.

2-P4: "although the interaction between N and P has been demonstrated to be crucial for the formation of cellular IBs, HMPV P can phase-separate independently in vitro refs 27,28."

Jargon: please rewrite.

3-P6. This is E. coli RNA non-specifically bound to recombinant N not viral RNA right? Should remind the lay reader...

4-P12: "positioned in the middle of the helix of N (Fig. 3D). "
Better to specify which helix it is. eg helix alpha5...

Reviewer #3 (Remarks to the Author):

Pneumoviruses are major human pathogens and understanding the molecular mechanisms they use to transcribe and replicate their genome is crucial to design new therapeutic approaches.

In this study the authors have solved by cryo-EM the structure of nucleocapsid-like particles of human metapneumovirus in complex or not with the C-terminal region of the phosphoprotein. This interaction is essential for the virus to tether the polymerase complex onto its encapsidated template. The region of the phosphoprotein involved in the interaction is very short, mainly disordered in solution, and interacts weakly with N. Solving the structure of such interaction by cryo-EM is a tour de force! As exemplified with the closely related respiratory syncytial virus, the structural information provided by this work could lead to the design of promising antiviral drugs.

The manuscript is well-written, the data presented are clear, and the claims are well-supported.

I only have a few suggestions that may improve the manuscript:

- When an interaction is described, alignments of the residues involved are shown but only for one of the two partners (i.e. in Fig. 2, it would be nice to see the conservation of the residues of the loop, and in Fig. 3 the conservation of the residues of N).
- Regarding the loop connecting N protomers, it would be great to discuss this rather 'new' interaction in light of the structures of the nucleocapsids of other mononegaviruses, especially paramyxoviruses.
- Since we have structural information on the L/P complex of HMPV, it would be nice to discuss more the interaction between the polymerase complex and the nucleocapsid. Can the monomer of PCT that binds L simultaneously interact with the nucleocapsid? What is the distance between the two binding sites (PCT-L and PCT-Nuc)?...
- In the result section, "black arrow in P disordered plot, Fig. 1A" should be replaced by "black arrow in P disordered plot, Fig. 3A".

Response to Reviewers

We would like to thank all 3 reviewers for their expert assessments of our manuscript. We believe that we have implemented changes to our paper in almost all instances where this was requested. Sections modified upon reviewers' requests are highlighted in the enclosed revised document. The revisions place our study better into the context of the literature, thereby highlighting the advance. Please find below point-by-point responses to the raised issues.

Reviewer #1 (Remarks to the Author):

This manuscript reports two structure studies, the N-RNA ring of HMPV and its complex with the C-terminus of phosphoprotein, by cryo-EM. The outcomes are very similar to those of other nonsegmented negative strand RNA (NNSR) viruses, including a closely related virus, RSV. Very little new information is revealed even though the results are confirmative of previous reports.

We respectfully disagree with the reviewer's assessment. The main source of criticism from reviewer 1 stems from the supposition that previous work on other non-segmented negative sense RNA viruses (nsNSVs) detracts from the significance of our work. The reviewer gives specific examples from previous studies from vesicular stomatitis virus (VSV), a member of the *Rhabdoviridae* which belongs to an evolutionary distinct nsNSV family, different from *Pneumoviridae*. In the following point-by-point responses we wish to highlight the substantial differences between nsNSV families, which illustrate that the individual study of their N-RNA/P interactions is important and necessary.

It is correct that nsNSVs share a common logic to their replicative cycles, however, the molecular mechanisms by which this is achieved vary between evolutionary remote families. For instance, in VSV the P protein possesses a folded N-binding domain that attaches directly to the C-terminal domain of the N protein (PMID 19571006). In contrast, in the family *Paramyxoviridae* (which includes e.g. mumps virus) the N proteins possess an additional large intrinsically disordered N_{tail} region (~170 residues in mumps) - N_{tail} is not present in VSV. The N_{tail} region utilizes a molecular recognition element to bind P, forming a P_{XD}-N_{MORE} complex (reviewed in PMID 34960734). The family *Pneumoviridae* again uses a different strategy, where a short motif of the disordered P C-terminus attaches directly to the N-terminal domain of N (this study). Taken together these examples demonstrate that viral families use different molecular strategies to achieve similar goals (attachment of L/P to template), under the involvement of differing binding modules. Extrapolating from only one virus family to all others is not sufficient to understand these interactions, and could potentially lead to wrong conclusions. In our study, we describe the N-RNA/P interaction in *Pneumoviridae* in greater detail than ever before – this interaction is different than in *Rhabdoviridae* or *Paramyxoviridae*. We therefore believe that our study constitutes a significant advance that helps us better understand a medically relevant virus.

However, we concede to reviewer 1 that we could have better placed our study into the context of the wider literature, specifically concerning *Rhabdoviridae* and *Paramyxoviridae*. Therefore, we have added a paragraph into our manuscript Introduction highlighting differences in N-RNA/P interactions throughout nsNSVs and citing relevant publications.

We address the comparison with RSV further below in this response letter.

The structure of N-RNA rings conformed that N protomers may adopt slightly altered interactions between them in the nucleocapsid, depending on the conformation of the nucleocapsid. Such variations are extensively reported previously, including the recent publications as the following:

Nat Commun. 2022 Oct 10;13(1):5980. doi: 10.1038/s41467-022-33664-4

Nat Commun. 2022 Aug 15;13(1):4802. doi: 10.1038/s41467-022-32223-1

which were omitted by the authors. The authors discussed the selected protomer conformation observed in their cryoEM structures of N-RNA rings with a small number of N protomers, which are just a few sampling points.

The papers referred to above analyse N protomer conformations in vesicular stomatitis virus (VSV). In these studies N conformations were assessed in the context of N subunits tightly packed against matrix (M) proteins in the bullet-shaped *Rhabdoviridae* particle. These studies are important and we have modified our manuscript to now cite them.

The short spiral N-RNA assembly is essentially the same as others.

We do not claim that the HMPV N-RNA spiral has a fundamentally different architecture to what has been seen in some other viruses. To clarify this, we have added citations to studies in which similar structures have been observed into the manuscript text.

For the complex between the N-RNA ring and the C-terminus of P, the structure is the same as previously reported for RSV.

We respectfully disagree with this assessment. We are not aware of any publication reporting assembled N-RNA bound to the P C-terminal region in RSV. The reviewer may be referring to the 2015 paper from Ouizougoun-Oubari et al., (PMID 26246564), which we cite in our manuscript. In that publication the authors made use of a single unassembled and truncated domain of N, in absence of RNA. They were also able to resolve only 2 single residues of the P peptide.

Even though the global location of those 2 residues is consistent, they do not align with the positions we observe for the P peptide in HMPV. Our structural analysis is therefore substantially more comprehensive and placed in a more relevant context. For these reasons we believe that our study is a significant advance beyond this previous work. We have added a supplementary figure (which we refer to in the Discussion) in which we supply a side-by-side comparison of the structures to highlight these points.

It is likely that other residues in the C-terminal region of P are also involved in the interactions with the N-RNA complex. As previously reported for other NNSR viruses, the C-terminal region of P selectively binds the nucleocapsid by interacting with two neighboring N protomers that are only present in the nucleocapsid assembly, although there is a major interaction area with one N protomer.

We are not aware of evidence showing that what the reviewer describes is also the case in *Pneumoviridae*. We feel that postulating this for pneumoviruses based solely on viruses from other

families would be highly speculative. In the pneumovirus field it is widely believed that a short stretch of residues at the C-terminal tip of P is required and sufficient for the N-RNA/P interaction. The example stated by the reviewer seems to refer to VSV, specifically the study: PMID 19571006. As detailed above, the interaction between P and N-RNA are fundamentally different between *Pneumoviridae* and *Rhabdoviridae*: *Pneumoviridae* P proteins possess a conditionally disordered C-terminus that attaches to the N-terminal domain of N. In contrast, in VSV the P protein possesses a folded N-RNA binding domain which attaches to the C-lobe of N (which is on the opposite side of the N molecule, compared to the equivalent site for pneumoviruses). There is no detectable homology between the C-terminal domain of *Pneumoviridae* and VSV P proteins.

The function of N binding by the C-terminus of P has been reported by this group in 2022, including minigenome assays, and IB formation, with introduction of mutations in P and N. The selection of mutated residues was slightly different in the 2022 publication, which is indicative of conformation-dependent interactions of P C-terminus with N. This proves that mutations selected in this paper are not more informative than those in the 2022 publication.

The reviewer correctly states that our mutagenesis experiments show conformation-dependent interactions. They also validate our structural observations of the involvement of N β HL in binding P. No mutations of N β HL were studied in the 2022 paper, as we did not possess any evidence of its involvement in binding. While N β HL is largely disordered in the unbound state, we observe increased ordering by binding of P. Taken together these experiments demonstrate that folding-upon-binding of a previously unexplored region is important in P recognition. We think that this is important data that goes beyond the 2022 publication.

Overall, the results included in this manuscript hardly add any new knowledge in this area, maybe just some small incremental pieces of new data.

As detailed in the beginning of the response letter, the reviewer bases this assessment largely on comparisons with studies on viruses from other viral families, citing VSV. Alignment of VSV P and HMPV P protein sequences (see the aligned sequences below in **Fig. 1**) yields a sequence identity of only 17% - essentially on the level of random similarities. This illustrates that P protein members of different viral families are sufficiently far removed as to render their individual study necessary. Moreover, our work goes substantially beyond previous research on respiratory syncytial virus, regarding comprehensiveness and conceptual findings. Finally, we add to the state of the field of a medically relevant virus and, especially, a binding site for an inhibitor in RSV. For the reasons stated here and above, we respectfully disagree with the reviewer and believe that our study is an advance of sufficient impact for Nature Communications.

```

HMPV_P      -MSFPEGKDILFMGNEAAKLAEFQKSLRKPSHKRSQSIGEKVNTVSETLELPTISRPT
VSV_P      MDNLTKVREYLKSYSRLDQAVGEIDEIEAQRAEKSNYLFE--DGVEEHTKPSYFQAAD
          .: : . : * . : . : : : * . . : : * : * * : . : . .
HMPV_P      KPTILSEPKLAWTDKGGAIKTEAKQTIKVMD-PIEEEEFTEKRVLPSSDGKTPAEKKLKP
VSV_P      DSDTESEPEIEDNQLYAQDPEAEQVEGFIQGPLDDYADEEVDVFTSDWKPP-----
          .. * : * : : . : * . . * : * . : : : * * : * * * *
HMPV_P      STNTKKKVSFTPNEPGKYTKLEKDALDLLSDNEEEDAESSILTFEERDTSSLSIEARLES
VSV_P      -----ELESDEHGKTLRLTSP--EGLSGEQKSQWLSTIKAVVQSAKYWNLAECTFEA
          .: : * * * * . * . : * : : : : : * : * : . : . * : * :
HMPV_P      IEEKLSM-----ILGLLRTLNIATAGPTAARDGIRDAMIGIREELIADIIEAKGKAAEM
VSV_P      SGEVIMKERQITPDVYKVTPVMNTHPS-----QSEAVSDVW--SLSKTSMT
          * : * . : . : . : : * : . . * : : * : . : * : :
HMPV_P      MEEEMNQRTKIGNGSVKLTEKAKELNKIVEDESTSGESEEEEEELKDTQENNQEDDIYQLI
VSV_P      FQPKKASLQPLTISLDELFSSRGEFISVGGDGRMSHKEAILLGLRYKKLYNQARVKYSL-
          : : : . : . : * . . * : : : * * : . * . : : * * *
HMPV_P      M
VSV_P      -

```

Fig. 1: **Sequence alignment of HMPV P and VSV P.** Sequences were aligned with MUSCLE (<https://www.ebi.ac.uk/Tools/msa/muscle/>). Accession codes: VSV P: P03520, HMPV P: Q8B9Q8.

Other concerns are as the following:

Abstract: The authors' statement "To define how L/P recognizes N-encapsidated RNA (N-RNA) we employed cryogenic electron microscopy" is not accurate. The results in this manuscript only defined the interactions between P and N-RNA. Interactions of L/P complex with N-RNA shall include interactions between P and N-RNA, and between L and N-RNA. The authors can only claim that they were to define the interactions between P and N-RNA.

While we are aware that direct binding of L to N has been captured in VSV particles (PMID 35476516), to the best of our knowledge there is no evidence of direct interactions between L residues and N residues in either HMPV or RSV, and the interaction is mediated exclusively by P. If the reviewer is aware of any studies showing direct interactions between L and N residues in HMPV or RSV we would be happy to add those references. However, we do agree that L was not present in our structural investigation and have therefore modified the sentence to a more cautious wording. We have also slightly modified the title of the manuscript to a further alleviate this concern.

"The nucleoprotein N is a promiscuous RNA-binding protein" is not a correct statement. N itself does not bind RNA per se. RNA is captured only if N can assemble into oligomers (see J Virol. 2008 Jan;82(2):674-82. doi: 10.1128/JVI.00935-07).

We have reworded the sentence accordingly.

"with the remainder being disordered." is not an accurate statement. P protein domains may assume different conformational states so they are not visible in the cryo-EM densities, but that does not mean that these domains are disordered.

In *Pneumoviridae*, only the L-interacting regions of P (including the tetramerization domain) can be considered as ordered. The remaining regions of P are either fully or conditionally disordered (see deep-learning based disorder metaprediction profile below, Fig. 2) . There is an abundance of data in the literature describing this, including from SAXS, MD, and NMR. See for instance PMID 28031463 or PMID 29093501. We are aware that in negative sense viruses from other families, such as VSV, P possess additional well-folded domains, but this is not the case here. We therefore stand by this statement.

Fig. 2: **Disorder plot for RSV P.** The plot was generated with metapredict (<https://metapredict.net/>). Uniprot accession code: P12579. The regions shaded in red are classified as disordered. The region from ~130 to 200 is found to interact with L.

"1) it tethers the L/P complex to the nucleocapsid while traversing the genome during transcription/replication," is not completely accurate. Is "it" P? P alone cannot traverse the genome.

We have modified the sentence for clarity.

"the structural mechanism of P attachment to N-RNA which retains substantial conformational flexibility in the bound state." What is "which"? P or N-RNA?

We agree with the reviewer's comment. This sentence has been reworded.

" HMPV N-RNA rings were expressed" is a sloppy statement. recombinant N was expressed and N assembles N-RNA rings by encapsidating host RNA.

We have modified the sentence for clarity.

"Mutagenesis of P308 to alanine resulted in a notable decrease of polymerase activity, while the R27A mutation nearly abrogated polymerase activity (Fig. 2F), suggesting that the N-N interaction via the 232-239 loop is important for successful N-RNA assembly." This conclusion is not sound. The mutations

have effects on minigenome assay, but there is no data on N-RNA assembly (see J Virol. 2008 Jan;82(2):674-82. doi: 10.1128/JVI.00935-07). The mutations could just cause conformational changes in N to interfere with interactions with L/P complex.

We have modified the section to specify the caveat noted by the reviewer.

"the C-terminal region of P (PCT)", "the 9 last residues of PCT (N-EDDIYQLIM-C)". The last 9 residues are not "the C-terminal region". It is just the C-terminus.

This is a question of definition. Strictly speaking, in a chemical sense, the C-terminus of a protein is the location of the terminal carboxyl group. However, some authors may refer to multiple residues at the end of the protein sequence as the 'C-terminus'. We have opted for the use of C-terminal region here, as we assessed the sequence in question to be too short as to be referred to as a 'domain', but also wanted to avoid the term C-terminus as this might be understood as just the last residue. As we specify the exact sequence in the quoted section, we do not believe that there is a risk of readers misconstruing the text and would prefer to keep the current nomenclature.

"These data indicate that the interaction between PCT and N-RNA is flexible and suggest that P has the ability to dynamically unbind and rebind on simulated timescales that are notably rapid (i.e. tens to hundreds of nanoseconds)." This is an overreaching statement. The MD simulation is only on 9 residues of P. It is most likely that there are more extensive interactions between the C-terminal region of P and N-RNA.

This is essentially the same argument as above, with the reviewer postulating a more extensive N-RNA/P binding surface, likely by comparison with *Rhabdoviridae*. As noted before, to the best of our knowledge there is no evidence for more extensive N-RNA/P interactions in *Pneumoviridae*. The reviewer referring to data from viruses with well-folded C-terminal domains of P which have no detectable homology to pneumovirus P.

"was not due to a defect of N expression or solubility, by purifying the individual N proteins expressed in bacteria (Fig. 5C)" But they did not check if these mutations disrupt proper N-RNA assembly.

We have modified the section to mention this possibility.

"Residues of HMPV P and N involved in the P-nucleocapsid interaction have been identified". Some residues, not all.

Again, this is the same avenue of criticism as before. The reviewer speculates that there should be a more extensive binding surface based on studies performed on members of other viral families, such as *Rhabdoviridae*. However, the aggregated literature on HMPV and RSV does not support this notion. In contrast, to the best of our knowledge, functional studies and structural studies (cited in the manuscript) in pneumoviruses support the notion of a short and conditionally disordered interaction motif as described in our work.

"The observed dynamics of N protomer orientations, even in the assembled state, may constitute a prerequisite for conformational changes facilitating access to the RNA genome during transcription by L/P". Overreaching conclusion.

We agree that this may be too speculative and have removed the statement.

Reviewer #2 (Remarks to the Author):

HMPV and RSV are major human viral pathogens causing serious respiratory infections. While a vaccine has been recently marketed for RSV prevention (<https://www.fda.gov/news-events/press-announcements/fda-approves-first-respiratory-syncytial-virus-rsv-vaccine>), no vaccine exists for HMPV. A better understanding of HMPV replication could help the design of antiviral therapies. Experimental structures are available for RSV and HMPV polymerase L/P complexes as well as for their ring-forming nucleocapsids. However, one key information lacking is how the CTD of the P phosphoprotein interacts with the nucleocapsid that wraps viral genomic RNA. To fill this gap, here, the authors report cryo-EM structures of recombinant HMPV N-RNA bound to PCT (N- RNA/P). They identify a conserved groove at the periphery of the N multimer to which P binds. This interface region is located between a helix and an extended loop of a beta-hairpin of N (NβHL). The authors use a minigenome experiments to validate the interaction as well as MD simulations of a complex between N-RNA/P to study the conformational dynamics of PCT. Overall, this is an important well-executed study. I only have a few comments.

We thank the reviewer for the encouraging assessment.

Major issues

1-P4: "The nucleoprotein N is a promiscuous RNA-binding protein with globular N-terminal and C-terminal domains (NTD and CTD, respectively)18,29. "

The introduction should spell out more clearly what has been published in terms of N-P interactions eg: ref 29 determined crystal structures of PNTD with recombinant NO (RNA free N protein). This is to highlight the really novel results reported in this ms for the lay reader.

We agree that this should be in the manuscript. We have added information on the previously published crystal structure of the complex between N₀ and the N-terminal domain of P.

2- Fig. 3F

HMPV-P: 288-DIYQLIM-294

Is this region truncated in RSV-P ?? One would expect conservation of this functionally important region between RSV-P and HMPV-P ? Please comment on this. Nonetheless, the authors rightly note

that the present work is in line with previous mapping of N-P interaction for HMPV reported in refs 27 and 28.

27. Thompson, R. E., Edmonds, K. & Dutch, R. E. Specific Residues in the C-Terminal Domain of the Human Metapneumovirus Phosphoprotein Are Indispensable for Formation of Viral Replication Centers and Regulation of the Function of the Viral Polymerase Complex. *J Virol* e0003023 (2023) doi:10.1128/jvi.00030-23.

28. Decool, H. et al. Characterization of the Interaction Domains between the Phosphoprotein and the Nucleoprotein of Human Metapneumovirus. *J Virol* 96, (2022).

The reviewer correctly recognizes that the HMPV P protein is significantly longer than RSV P (294 vs. 241 residues, respectively). The C-terminal regions of the two proteins are also somewhat divergent. For comparison, we have now added the sequence of last residues of RSV P to Fig. 3 of the manuscript. The panel shows that, although both viruses belong to *Pneumoviridae*, there is a marked sequence divergence in this region, indicating that the specific interactions with N-RNA may very well vary between viruses. We now also note this in the text.

3- One would expect the interaction between PCTD and N to destabilize the interaction between N and RNA to allow threading of RNA into the polymerase for replication/transcription. This does not seem to be supported by the manuscript. This should be discussed in terms of a revised model of replication/transcription. Maybe using a figure or a video capturing the key results of the present work.

This is an excellent point and a major question in the field. RNA synthesis in nsNSVs is initiated either at the leader sequence at the 3'-end of genomes or the trailer sequence at the 3'-end of antigenomes. As correctly implied by the reviewer, the structure of the polymerase sterically cannot accommodate N-RNA. This means that the viral RNA must be dissociated from N such that an initiation complex can form.

To further test the effect of P-binding on the packaged RNA, we have carried out an MD simulation of N-RNA in absence of P. We observe that the RMSD of the bound RNA is slightly higher in the N-RNA/P_{CT} simulations than with only N-RNA. However, the effect is rather small and the number of hydrogen bonds between RNA and N is not significantly changed between N-RNA/P_{CT} and N-RNA in the course of the simulations. These data are now included in the manuscript. The simulations and the cryo-EM data indicate that binding of the C-terminal region of P to N-RNA alone does not sufficiently destabilize the bound RNA, and additional factors may be required at the viral promoter sites.

One possible mechanism may involve the free CTD-arm of the N protein located at the 3'-end of the packaged RNA. Previous research has indicated that this region, located at the C-terminal end of the nucleoprotein, possesses some inherent affinity to the RNA binding groove and can insert itself into it (PMID: 26880565). Perhaps it is the CTD-arm of N that assists in dislodging the RNA at the 3'-end by competing for the RNA-binding groove and thus prepares the formation of an initiation complex. We have added this idea as a tentative suggestion to the discussion, have refrained from making a separate figure, however, due to its speculative nature.

4- R132 from N-HPMV appears to play a key role for the interaction with PCT, how evolutionary conserved is R132?

Indeed, in line with its central role in interacting with P, R132 is highly conserved in *Pneumoviridae*. We have added a MSA of N as a panel in the supplementary material and now note the observation in the manuscript text.

5- Fig. 1 A: Elution profile from superose 6. Some MW standards should be added to this panel for consistency with the other panels.

Agreed. We have added a SEC standard into the plot.

6- The two supplementary videos are not very informative. Is it possible to improve them in terms of content (eg adding labels, arrows, captions). This is an important work that should be well presented and informative videos could help.

Agreed. We have added labels and arrows to the supplementary videos to improve their presentation.

Minor issues

1-P3 Currently, there are no licensed vaccines or specific therapeutics for the treatment of HMPV infections. There is now a licensed vaccine for the closely related RSV.

We now note this in the Introduction.

2-P4: "although the interaction between N and P has been demonstrated to be crucial for the formation of cellular IBs, HMPV P can phase-separate independently in vitro refs 27,28." Jargon: please rewrite.

Agreed, we have rephrased this.

3-P6. This is E. coli RNA non-specifically bound to recombinant N not viral RNA right? Should remind the lay reader...

Yes, we now specify this here explicitly.

4-P12: "positioned in the middle of the helix of N (Fig. 3D). "

Better to specify which helix it is. eg helix alpha5...

Agreed, we now refer to this helix as α_{12} , in accordance with previously established nomenclature (PMID 19965480).

Reviewer #3 (Remarks to the Author):

Pneumoviruses are major human pathogens and understanding the molecular mechanisms they use to transcribe and replicate their genome is crucial to design new therapeutic approaches.

In this study the authors have solved by cryo-EM the structure of nucleocapsid-like particles of human metapneumovirus in complex or not with the C-terminal region of the phosphoprotein. This interaction is essential for the virus to tether the polymerase complex onto its encapsidated template. The region of the phosphoprotein involved in the interaction is very short, mainly disordered in solution, and interacts weakly with N. Solving the structure of such interaction by cryo-EM is a tour de force! As exemplified with the closely related respiratory syncytial virus, the structural information provided by this work could lead to the design of promising antiviral drugs.

The manuscript is well-written, the data presented are clear, and the claims are well-supported.

We thank the reviewer for the kind and supportive assessment.

I only have a few suggestions that may improve the manuscript:

- When an interaction is described, alignments of the residues involved are shown but only for one of the two partners (i.e. in Fig. 2, it would be nice to see the conservation of the residues of the loop, and in Fig. 3 the conservation of the residues of N).

Agreed. We have added a panels showing MSAs of the aforementioned regions of N to Fig. 2 and Fig.3 of the manuscript and refer to the panels in the text.

- Regarding the loop connecting N protomers, it would be great to discuss this rather 'new' interaction in light of the structures of the nucleocapsids of other mononegaviruses, especially paramyxoviruses.

Agreed, we have added and refer to an additional supplementary figure in the manuscript where we make this side-by-side comparison. We observe that in *Paramyxoviridae* an equivalent loop exists. However, it is less tightly associated with its neighbouring N-RNA protomers. To quantify this we used the PISA server (<https://www.ebi.ac.uk/pdbe/pisa/>) and calculated buried surface areas involving the interface between loop and neighbouring N-RNA subunit: while the interface in HMPV covers $\sim 680 \text{ \AA}^2$, in measles it is only $\sim 395 \text{ \AA}^2$. Based on these analyses we suggest that the role of the loop in facilitating N-N lateral interaction is more pronounced in pneumoviruses.

- Since we have structural information on the L/P complex of HMPV, it would be nice to discuss more the interaction between the polymerase complex and the nucleocapsid. Can the monomer of PCT that binds L simultaneously interact with the nucleocapsid? What is the distance between the two binding sites (PCT-L and PCT-Nuc)?...

This is an excellent point. As the reviewer correctly mentions, and as stated in the manuscript, the tetrameric P protein binds L in an asymmetric manner, placing three copies of P on one side of the polymerase and a fourth copy of P winds around the opposing side. In RSV, the C-terminus of the fourth copy of P is fully bound to the L polymerase, rendering it unavailable for attachment to N (PMID 31953395). This indicates, that in RSV at most three of the four copies of tetrameric P could engage the nucleocapsid.

The situation is different in HMPV, which possesses a longer P protein (241 residues in RSV, 294 residues in HMPV). In HMPV, the single copy of P on the opposing side of the polymerase binds L only up to residue 266. This means that there is a spacer of around 20 residues until the N-RNA binding motif starts. In theory, this spacing may be of sufficient length (theoretical maximal length of a fully outstretched peptide = 3.8 Å per residue) to enable this P copy to also contact the nucleocapsid, in contrast to RSV. However, considering the homology to RSV, we think it is more likely that this is not the case and only 3 P copies attach to N-RNA, consistent with RSV.

The remaining three copies of HMPV P bind to L up to residue 236. This implies a sizable linker of around 50 residues distance to the N-RNA binding site. With a disordered linker of such considerable length it is feasible that L/P attaches to N-RNA protomers at a substantial distance, possibly spanning consecutive turns on the helical nucleocapsid.

However, in absence of structural data of the full L/P-nucleocapsid complex, the discussion above remains highly speculative. We believe that structural data, even at fairly low resolution, of nucleocapsid-bound L/P is urgently needed to better understand transcription. For instance, given a simple distance measurement of transcribing L/P to nucleocapsid, we might be able to draw conclusions on how many P copies interact with N and how much of the RNA genome is dissociated from the nucleocapsid.

We have now incorporated some of these ideas into a new paragraph of the discussion of the manuscript, as requested by the reviewer.

- In the result section, “black arrow in P disordered plot, Fig. 1A” should be replaced by “black arrow in P disordered plot, Fig. 3A”.

Many thanks for catching this mistake - we have corrected it.

REVIEWERS' COMMENTS

Reviewer #1 (Remarks to the Author):

The authors revised portion of the manuscript in response to some of the comments by this reviewer. On the other hand, the authors wrote extensive arguments, which does not resolve the concerns of this reviewer.

Reviewer #2 (Remarks to the Author):

The authors have addressed well the comments.

Reviewer #3 (Remarks to the Author):

I thank the authors for taking my remarks into consideration.
I agree with the publication of the revised manuscript.

Response to Reviewers

We would like to thank all 3 reviewers for their final assessments of our revised manuscript. We believe that the manuscript overall is significantly strengthened by the revisions.

Reviewer #1 (Remarks to the Author):

The authors revised portion of the manuscript in response to some of the comments by this reviewer. On the other hand, the authors wrote extensive arguments, which does not resolve the concerns of this reviewer.

We thank reviewer 3 for their feedback. While we hoped to have provided a convincing argument on what constitutes sufficient impact, we will have to agree to disagree on this matter. We do appreciate the points made by the reviewer which helped improve the manuscript.

Reviewer #2 (Remarks to the Author):

The authors have addressed well the comments.

Reviewer #3 (Remarks to the Author):

I thank the authors for taking my remarks into consideration.
I agree with the publication of the revised manuscript.

Many thanks to reviewers 2 and 3 for their helpful suggestions and kind feedback.